# Theoretical tool bridging cell polarities with development of robust morphologies

**Silas Boye Nissen[1,2], Steven Rønhild[1], Ala Trusina[1,2]\*, Kim Sneppen[1]\***

[1]Center for Models of Life, Niels Bohr Institute, University of Copenhagen, Copenhagen, Denmark; [2]StemPhys, Niels Bohr Institute, University of Copenhagen, Copenhagen, Denmark

**Abstract** Despite continual renewal and damages, a multicellular organism is able to maintain its complex morphology. How is this stability compatible with the complexity and diversity of living forms? Looking for answers at protein level may be limiting as diverging protein sequences can result in similar morphologies. Inspired by the progressive role of apical-basal and planar cell polarity in development, we propose that stability, complexity, and diversity are emergent properties in populations of proliferating polarized cells. We support our hypothesis by a theoretical approach, developed to effectively capture both types of polar cell adhesions. When applied to specific cases of development – gastrulation and the origins of folds and tubes – our theoretical tool suggests experimentally testable predictions pointing to the strength of polar adhesion, restricted directions of cell polarities, and the rate of cell proliferation to be major determinants of morphological diversity and stability.

DOI: https://doi.org/10.7554/eLife.38407.001

**\*For correspondence:**
trusina@nbi.ku.dk (AT);
sneppen@nbi.dk (KS)

**Competing interests:** The authors declare that no competing interests exist.

## Introduction

Multicellular organisms are amazing in their ability to maintain complex morphology in face of continuous cell renewal and damages. Adult salamander can regenerate entire limbs (*Eguchi et al., 2011*), and, during development, some regions can maintain patterning when moved to different parts of an embryo or if the size is varied (*Lyons et al., 2012*). Given the vast complexity and diversity of living shapes, how can we reconcile the robustness to perturbations with flexibility to diversify? While undoubtedly the end result is encoded in the DNA and protein networks, looking for an answer at this level is challenging. Examples of phenotypic plasticity (*Libby and Rainey, 2011*), convergent evolution, and contrasting rates of morphological and protein evolution (*Cherry et al., 1979*) show that morphological similarity may not couple to the protein sequence similarity (*Gould, 1970*). Inspired by the unfolding of morphological complexity in development, we propose that cellular polarity may be the key for reconciling complexity, robustness, and diversity of organismal morphologies.

During early development, the increase in morphological complexity coincides with the progressive polarization of cells – first apical-basal (AB) polarity and then planar cell polarity (PCP) (*Müller and Bossinger, 2003*; *Roignot et al., 2013*; *Andrew and Ewald, 2010*; *Li and Bowerman, 2010*). This theme is ubiquitous across vertebrates and invertebrates (*Figure 1*): starting from the single fertilized egg cell, first the morula formed by non-polarized cells turns into the blastula – a hollow sphere of cells with AB polarity. Then, as cells acquire additional PCP, primary head-tail axis forms and elongates during gastrulation and neurulation (*Loh et al., 2016*). Because of the optical transparency, these stages are particularly prominent in sea urchin. At the morula stage, a lumen in the center is formed and is gradually expanding as cells proliferate and rearrange into the hollow

**eLife digest** Cells have the power to organise themselves to form complex and stable structures, whether it is to create a fully shaped baby from a single egg, or to allow adult salamanders to grow a new limb after losing a leg. This ability has been scrutinised at many different levels. For example, researchers have looked at the chemical messages exchanged by cells, or they have recorded the different shapes an embryo goes through during development. However, it is still difficult to reconcile the information from these approaches into a description that makes sense at multiple scales.

When an embryo develops, sheets of cells fold and unfold to create complex 3D shapes, like the tubes that make our lungs. Moulding sheets into tubes relies on interactions between cells that are not the same in all directions. In fact, two types of asymmetry (or polarity) guide these interactions. Apical-basal polarity runs across a sheet of cells, which means that the top surface of the sheet differs from the bottom. Planar cell polarity runs along the sheet and distinguishes one end from the other. For instance, apical-basal polarity marks the inner and outer surfaces of our skin, while planar cell polarity controls the direction in which our hair grows.

Nissen et al. set out to investigate how these polarities help cells in an embryo organise themselves to form complicated folds and tubes. To do this, simple mathematical representations of both apical-basal and planar cell polarities were designed. The representations were then combined to create computer simulations of groups of cells as these divide and interact with each other.

Simulations of 'cells' with only apical-basal polarity were able to generate different shapes in the 'tissues' produced, including many found in living organisms. External conditions, such as how cells were arranged to start with, determined the resulting shape. With both apical-basal and planar cell polarities, the simulations reproduced an important change that occurs during early development. They also replicated how the tubes that transport nutrients and oxygen form.

These results show that simple properties of individual cells, such as polarities, can produce different shapes in developing tissues and organs, without the need for a complicated overarching program. Abnormal changes in cell polarity are also associated with diseases such as cancer. The mathematical model developed by Nissen et al. could therefore be a useful tool to study these events.

DOI: https://doi.org/10.7554/eLife.38407.002

sphere. Next, during gastrulation, a group of cells invaginate and rearrange into a tube that narrows and elongates primarily by cell rearrangement and convergent extension movements (*Martik and McClay, 2017*). The tube then merges with the sphere at the side opposite to invagination, and as a result, the sphere transforms into a torus. Emerging data suggest that PCP drives both invagination and tube elongation (*Nishimura et al., 2012*; *Croce et al., 2006*; *Long et al., 2015*) – a recurring theme across species.

Mutations in PCP pathways produce shorter and wider tubes (*Ochoa-Espinosa et al., 2012*; *Saburi et al., 2008*; *Kunimoto et al., 2017*), somites (*Song et al., 2010*), and embryos (*Gong et al., 2004*). Formation and elongation of the tubes can proceed without cell division and cell death by cells rearranging along the tube's axis, termed convergent extension (CE) (*Andrew and Ewald, 2010*; *Tanimizu et al., 2009*; *Martik and McClay, 2017*). While the importance of PCP in gastrulation and tubulogenesis is well established (*Andrew and Ewald, 2010*; *Tanimizu et al., 2009*; *Martik and McClay, 2017*; *Kunimoto et al., 2017*), it is unclear how polarity may control tube morphology.

The bulk-lumens-folds/tubes transition seen across animal species in early embryogenesis, is also a key feature of the later organ formation. The early stages of organogenesis in the liver, kidney, brain, gut, and pancreas are apparently so robust, that they can be recapitulated in vitro, allowing for advanced quantification and manipulation (*Little, 2017*). The case of pancreatic organoids is interesting as it illustrates an increase of morphological complexity from spheres to folds. Cells in in vitro pancreatic organoids first grow as a bulk and later acquire AB polarity and develop lumens. Depending on the growth conditions, organoids develop into a hollow sphere or acquire a complex folded shape (*Greggio et al., 2013*). It is currently unknown what drives the transition from sphere

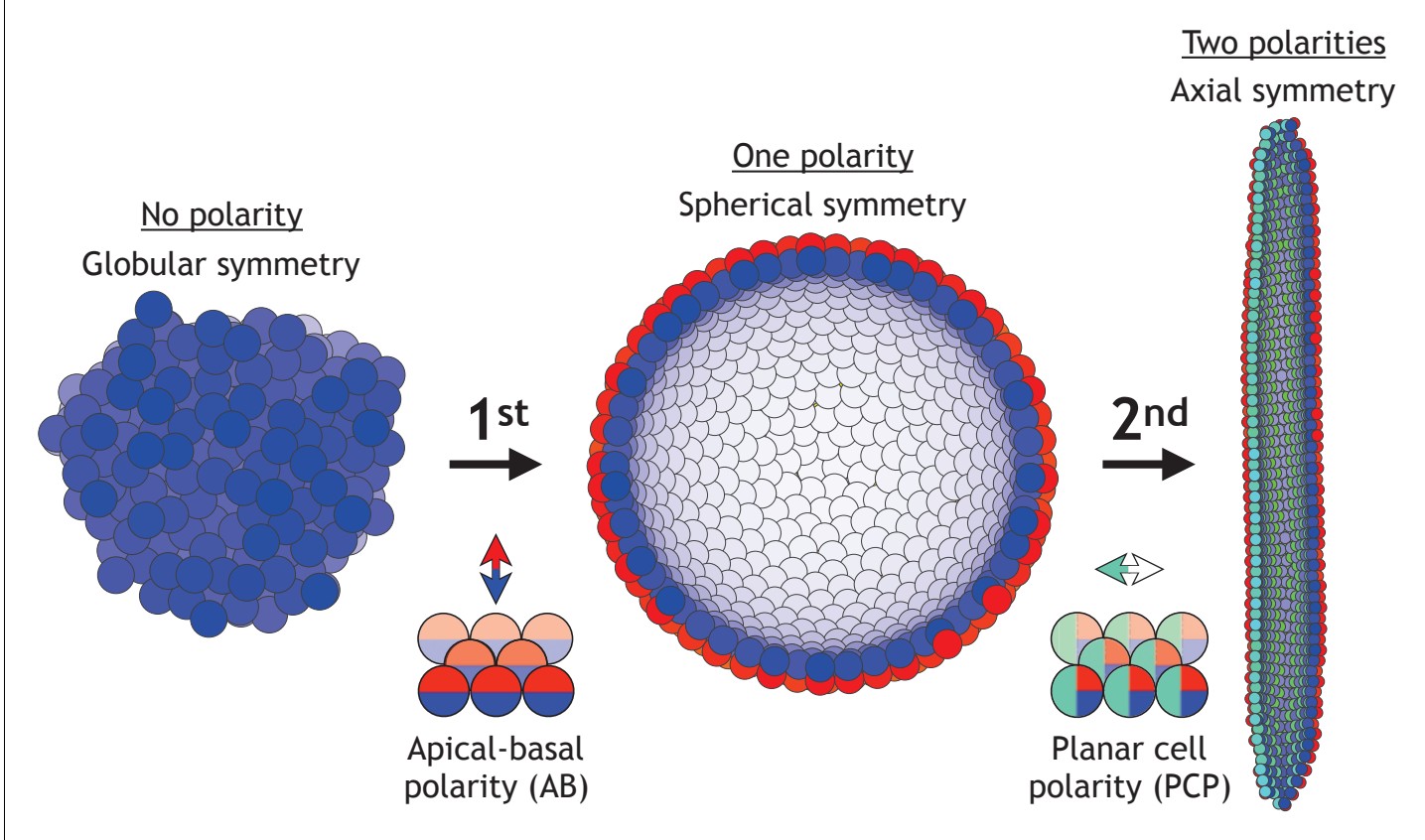

**Figure 1.** Two symmetry-breaking events, gain of apical-basal (AB) polarity and planar cell polarity (PCP), on cellular level coincide with the appearance of a rich set of morphologies. Starting from an aggregate of non-polarized cells (globular symmetry), individual cells can gain AB polarity and form one or multiple lumens (spherical symmetry). Additional, gain of PCP allows for tube formation (axial symmetry). Complex morphologies can be formed by combining cells with none, one, or two polarities. In *Figure 1—figure supplement 1*, we schematically illustrate how existing models capture different elements of development.

DOI: https://doi.org/10.7554/eLife.38407.003

The following figure supplement is available for figure 1:

**Figure supplement 1.** Overview of the existing literature on models addressing specific developmental events discussed in our work.
DOI: https://doi.org/10.7554/eLife.38407.004

to folded state; the two possible hypotheses are rapid proliferation or physical pressure from growing into a stiff matrigel.

Is the apparent link between cellular polarity and morphological complexity accidental? Or, could it be that morphological transitions, stability, and diversity are emergent features in a population of proliferating polarized cells? If true, can we identify what drives the transition from lumens to folds and tubes? Why are these stable? Can we predict what controls fold depth, and tube length and width? To answer these questions, we lack a unified approach that could bridge polar interactions between single cells to the global features emerging on the scale of thousands of cells in 3D.

Starting with D'Arcy Thompson's seminal contribution (*Thompson, 1942*), quantitative models aided in understanding specific morphogenetic events (*Figure 1—figure supplement 1*). Among these are *invagination* (*Odell et al., 1981*; *Rauzi et al., 2015*; *Polyakov et al., 2014*; *Hočevar Brezavšček et al., 2012*), primitive streak formation (*Newman, 2008*), convergent extension (*Collinet et al., 2015*; *Belmonte et al., 2016*), epithelial folding (*Buske et al., 2012*; *Osterfield et al., 2013*; *Monier et al., 2015*; *Murisic et al., 2015*), emergence of global PCP alignment from local cell–cell coupling (*Amonlirdviman et al., 2005*; *Le Garrec et al., 2006*; *Burak and Shraiman, 2009*), origins of *tubulogenesis* (*Engelberg et al., 2008*), and recently statistical properties of branching morphogenesis (*Hannezo et al., 2017*). However, they are often on either of the

two ends of the spectra: those modeling single cells explicitly, often rely on vertex-based approaches and are limited to dozens of cells (*Alt et al., 2017*; *Misra et al., 2016*; *Aigouy et al., 2010*; *Le Garrec et al., 2006*). To capture the large features spanning thousands of cells, one typically turns to elastic models where AB polarity is implicit and epithelia is presented as a 2D elastic sheet (*Hannezo et al., 2014*; *Etournay et al., 2015*; *Hufnagel et al., 2007*; *Nagai and Honda, 2009*; *Aliee et al., 2012*; *Nagai and Honda, 2001*).

We developed a theoretical approach that, with only a few parameters, bridges cellular and organ scales by integrating both types of polarity. A main difference to earlier approaches is that a cell's movement is coupled to how its AB polarity and PCP are oriented relative to each other and relative to neighbor cell polarities. In other words, in our approach, the adhesion strength between neighbor cells is modulated by the orientation of their polarities. We find that polarity enables complex shapes robust to noise but sensitive to changes in initial and boundary constraints, thus supporting that morphological stability and diversity are emergent properties of polarized cell populations. Lumens, folds, and stable tubes emerge as a result of energy minimization. We make testable predictions on morphological transitions in pancreatic organoids, tubulogenesis, and sea urchin gastrulation. Our approach illustrates the evolutionary flexibility in the regulatory proteins and networks, and suggests that despite differences in proteins between organisms, the same core principles may apply.

## Model

There are three key elements that allow us to bridge the scale from cellular level to macroscopic stable morphologies.

## (1) Cells are approximated by point particles

Cell–cell adhesion is modeled by repulsive and attractive forces acting between cell centers. This allows a substantially gain in computation time compared to vertex-based models where cell surface adhesion is explicitly considered (*Alt et al., 2017*). The potential for pairwise interaction between two interacting neighbors, $i$ and $j$, separated by distance $r_{ij}$ is

$$V_{ij} = e^{-r_{ij}} - S\,e^{-r_{ij}/\beta}, \tag{1}$$

where the first term corresponds to repulsion, and the second term to attraction (see *Figure 2A*). For a pair of non-polar cells, the strength of attraction $S = 1$. $\beta > 1$ is the parameter that sets how much longer the attraction range is compared to repulsion. We set $\beta = 5$ throughout the paper, but our results and conclusions are consistent for smaller $\beta$. The main results are also not sensitive to the exact choice of the potential, thus for example the higher power in the exponential,

$$V_{ij} = e^{-(r_{ij})^4} - S\,e^{-(r_{ij}/\beta)^4}, \tag{2}$$

give qualitatively similar results (see *Figure 2—figure supplement 1A*). The potential energy of a cell is the sum of pairwise neighbor interactions

$$V_i = \sum_j V_{ij}. \tag{3}$$

## (2) Cells interact with (a subset of Voronoi) neighbors

Interacting neighbors of cell $i$ are selected from a subset of cells sharing a Voronoi surface. The subset is limited to the nearest neighbors $j$ which are closest to the midpoint between $i$ and $j$ (*Figure 2B–C*). This constrain effectively corrects for the finite volume associated with point particles and assures that two cells will not interact if the line of sight between their centers is separated by a surface of a third cell. Without this constraint, the macroscopic morphologies collapse. However, our results are robust to replacing the line of sight constraint with full Voronoi and a cut-off distance for attraction force (*Figure 2—figure supplement 1B*).

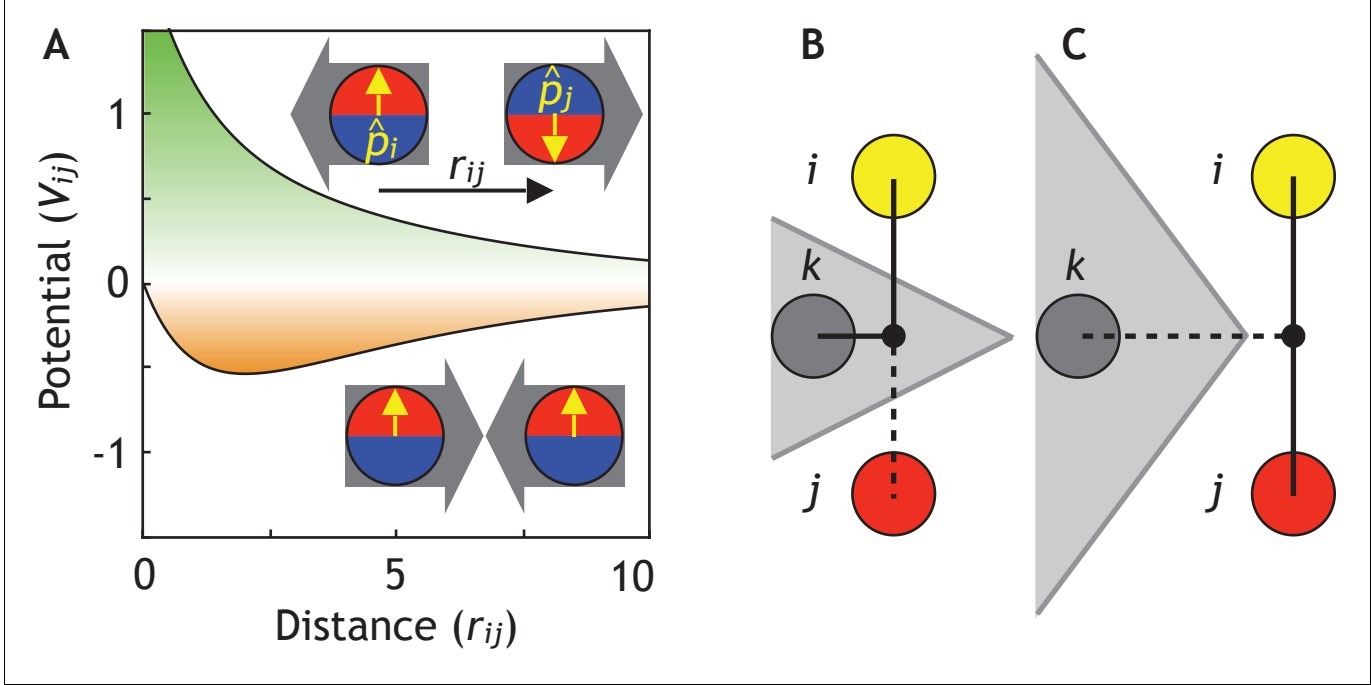

**Figure 2.** Cells are modeled as interacting particles with a polarity-dependent potential. (**A**) Potential between two interacting cells with apical-basal polarity (see *Equation 6*). Cells repulse when polarities are antiparallel (top/green part) and attract when they are parallel (orange/bottom part). (**B–C**) Two cells interact only if no other cells block the line of sight between them. (**B**) Cell *i* and *j* do not interact if *ij*'s midpoint (black dot) is inside of the Voronoi diagram for cell *k* (shaded in grey). (**C**) Cell *i* and *j* interact because cell *k* is further away than the distance *ij*/2 and *ij*'s midpoint therefore lie outside of cell *k*'s Voronoi diagram. In the related *Figure 2—figure supplement 1A*–D, we test the sensitivity of our model to the details of the potential and neighborhood assignments. In *Figure 2—figure supplement 2*, we relate changes in cell shapes to the model components, and in *Figure 2—figure supplement 3* (*Figure 2—video 1*), we illustrate how altering polarity affects the dynamics of the systems with two and six cells.
DOI: https://doi.org/10.7554/eLife.38407.005

The following video and figure supplements are available for figure 2:

**Figure supplement 1.** Dependence on the shape of the physical potential, the interaction partners, and noise.
DOI: https://doi.org/10.7554/eLife.38407.006

**Figure supplement 2.** Changes in cell shapes may reorient apical-basal (AB) polarity.
DOI: https://doi.org/10.7554/eLife.38407.007

**Figure supplement 3.** Examples of simple systems consisting of only two or six cells (see also *Figure 2—video 1*).
DOI: https://doi.org/10.7554/eLife.38407.008

**Figure 2—video 1.** Dynamics for two and six interacting cells.
DOI: https://doi.org/10.7554/eLife.38407.009

### (3) Cell–cell adhesion depends on the orientation of polarity

To capture directional adhesion, we set the strength of attraction, *S*, to be dependent on the relative orientation of the polarities in each of the cells. We assume that AB polarity and PCP are orthogonal, and that the polarities of one cell align with the polarities of its neighbor cells. Mathematically, we introduce unit vectors, $\hat{p}_i$ and $\hat{p}_j$ representing AB polarity, and $\hat{q}_i$ and $\hat{q}_j$ representing PCP for cell *i* and *j*, respectively. We set

$$S = \lambda_1 S_1 + \lambda_2 S_2 + \lambda_3 S_3 \tag{4}$$

where $\lambda_1$, $\lambda_2$, and $\lambda_3$ are the strengths of the different polarity terms. We require that

$$\lambda_1 + \lambda_2 + \lambda_3 = 1, \tag{5}$$

to satisfy the constraint that perfectly aligned cells always have a steady state distance of 2 cell radii (for $\beta$ = 5). To capture that in an epithelial sheet, AB polarities align parallel to each other and

tight adherens junctions form in the plane perpendicular to AB polarity, we introduce the quadruple product

$$S_1 = (\hat{p}_i \times \hat{r}_{ij}) \cdot (\hat{p}_j \times \hat{r}_{ij}). \tag{6}$$

This makes two interacting cells with AB polarity maximally attracted ($S_1 = 1$) if the two apical sides are next to each other. On the other hand, if apical side of one cell is next to the basal side of another cell, the two cells will be maximally repulsing ($S_1 = -1$), see *Figure 2A*.

In case of planar polarization, we define

$$S_2 = (\hat{p}_i \times \hat{q}_i) \cdot (\hat{p}_j \times \hat{q}_j), \tag{7}$$

which makes the attraction maximal if the PCP of two cells are parallel to each other and perpendicular to their AB polarities. In addition, we assume that similarly to AB polarity, two cells with PCP are maximally attracted if their PCP are parallel and cells have the same kind of pole (e.g. Vangl-enriched) next to each other,

$$S_3 = (\hat{q}_i \times \hat{r}_{ij}) \cdot (\hat{q}_j \times \hat{r}_{ij}). \tag{8}$$

We later show that this assumption makes neighbor exchange on a sheet possible and results in CE. However, unlike with tight junctions, preferred directional adhesion with PCP is not as well established. Cells adhere to each other by membrane proteins assembled in adherens junctions just below the apical surface. Both proteins regulating adherens junctions, for example Smash (*Beati et al., 2018*), as well as adherence proteins forming adherens junctions can be planar polarized, for example Bazooka, E-cadherins (*Simões et al., 2010*; *Tamada et al., 2017*; *Levayer and Lecuit, 2013*; *Warrington et al., 2013*; *Aigouy and Le Bivic, 2016*). These indirectly support our assumption of anisotropic, planar polarized adhesion.

The motion of the cells and their polarities are calculated assuming overdamped dynamics

$$\frac{d\bar{r}_i}{dt} = -\frac{dV_i}{d\bar{r}_i} + \eta, \tag{9}$$

$$\frac{d\bar{p}_i}{dt} = -\frac{dV_i}{d\bar{p}_i} + \eta, \tag{10}$$

and

$$\frac{d\bar{q}_i}{dt} = -\frac{dV_i}{d\bar{q}_i} + \eta, \tag{11}$$

where the $\bar{p}_i$ and $\bar{q}_i$ differentiation takes into account the rotation of polarity vectors, and $\eta$ is a random uncorrelated Gaussian noise. In practice, we implemented the model in a MATLAB script (available in the Materials and methods section), where we use the Euler method. We perform the differentiation along the polarity by differentiating along all three cartesian coordinates (see Model details in the Materials and methods section). After each time step, we normalize the updated polarity vectors. The above differentiation does not include the change in partners when neighborhood changes. This is treated as a non-equilibrium step where potential energy can increase (*Equation 3*). Biologically, this is similar to cells spending biochemical energy as they rearrange their neighborhood.

The point particle approximation has been utilized earlier for modeling non-polar cell adhesion in early blastocyst (*Krupinski et al., 2011*), slug formation in amebae (*Dallon and Othmer, 2004*), and PCP organization in primitive streak formation (*Newman, 2008*). The main novelty of our approach is the dynamical coupling of cell positions and polarity orientations (*Equation 6–8*).

## Model implications

One of the implications of the coupling between position and polarity is that in a sheet of cells, turning AB polarity in one cell will cause a force on its neighbors. In case of two cells (*Figure 2—figure supplement 3* and *Figure 2—video 1*), the pair relaxes the imposed stress by rotating both the polarity and their positions. In biological terms, turning AB polarity in one cell (e.g. by apical

constriction, illustrated in *Figure 2—figure supplement 2*) of an epithelial sheet will induce bending of the sheet as is the case with bottle cells in invagination.

The present formulation of PCP has several implications. First, we restrict the effects of PCP to directed (anisotropic) cell–cell adhesion and do not consider its other possible roles, in for example asymmetric cell differentiation, thus primarily focusing on its role on CE. Second, in our current formulation, AB polarity and PCP influence each other's orientation on equal footing (*Equation 7*). PCP, however, is typically constrained to the apical plane and thus is expected not to influence the orientation of AB polarity. Disabling PCP's effect on AB polarity (see Materials and methods) does not influence our main results on tube formation and gastrulation. However, the symmetry in polarities is appealing for its simplicity and is indirectly supported by the following experimental observations: First, cells can acquire PCP without AB polarity present (*Baer et al., 2009*; *Zorn and Wells, 2009*). Second, proteins required for AB polarity can be planar-polarized (*Warrington et al., 2013*; *Aigouy and Le Bivic, 2016*; *Beati et al., 2018*; *Choi and Sokol, 2009*; *Dollar et al., 2005*; *Kaplan and Tolwinski, 2010*). Third, changes in cell shapes during invagination (e.g. sliding of adherens junctions and formation of bottle cells) are regulated by PCP in neural tube closure (*Ossipova et al., 2014*; *Nishimura et al., 2012*; *Kinoshita et al., 2008*), gastrulation in *C. elegans* (*Lee et al., 2006*), sea urchin (*Croce et al., 2006*), and *Xenopus* (*Choi and Sokol, 2009*). These changes in cell shape effectively reorient AB polarity (*Figure 2—figure supplement 2*).

## Results

We have recently introduced effective representation of AB polarity, and showed that it is sufficient for capturing spherical trophectoderm in the early blastocyst (*Nissen et al., 2017*). Expanding on that work, we here explore how AB polarity supports diverse yet stable and complex morphologies.

### Stable complex shapes emerge from randomly polarized cell aggregates

Adult organismal shapes are stable over long time, maintaining sizes and relative positions of lumens and folds, despite continual local damages and cell renewal. To test if cellular polarity could enable such stability in time and to random local perturbations, we first performed a series of tests with AB polarized cells (*Figure 3* and *Figure 3—video 1*).

When starting a bulk of cells with AB polarities pointing randomly, an initial rapid expansion (*Figure 3A–C*) stabilizes into a complex morphology of interconnected channels (*Figure 3C–E*). The shape remains unchanged for at least 10 times longer than the initial expansion (*Figure 3C–E*). The stability of the shape is illustrated by the time evolution of the average energy per cell (*Figure 3F*) that after an initial fast drop converges to a constant value. As expected, this value is higher than the energy of a hollow sphere (yellow dot in *Figure 3F*) – a configuration obtained if we start with radially, instead of random, polarized cells and preserve radial polarization at all times. The observed behavior is not sensitive to the shape of the potential (*Figure 2—figure supplement 1A*) but is sensitive to how the neighborhood is defined (*Figure 2—figure supplement 1B–D*). Rerunning the simulation in *Figure 3* with different initial conditions results in a different stable shape (*Figure 2—figure supplement 1E* and *Figure 3—figure supplement 1*).

The macroscopic features of the shapes are robust to noise (*Figure 2—figure supplement 1E–F* and *Figure 3—figure supplement 1*). While the shapes emerging under high and low noise are not identical, the relative position and sizes of the majority of channels and lumens are preserved. The changes caused by noise stem from perturbations during initial expansion stage. If the same level of noise is applied after the system reached stable state, after time $t = 10000$, noise does not cause any major macroscopic changes (*Figure 2—figure supplement 1G*). The obtained shapes have self-sealing features, as an initial cut and unwrapping of a section of a surface refolds and seals back into the original morphology (*Figure 3—figure supplement 2A–C*). Furthermore, the shapes (*Figure 3D–E*) are also robust to overall growth (*Figure 3—figure supplement 2D–F*) retaining the same macroscopic features, just scaled to a larger size. Robustness to noise and cell proliferation further support the link between polarity and stability of morphologies, for example organ shapes, as they expand from infant to adult.

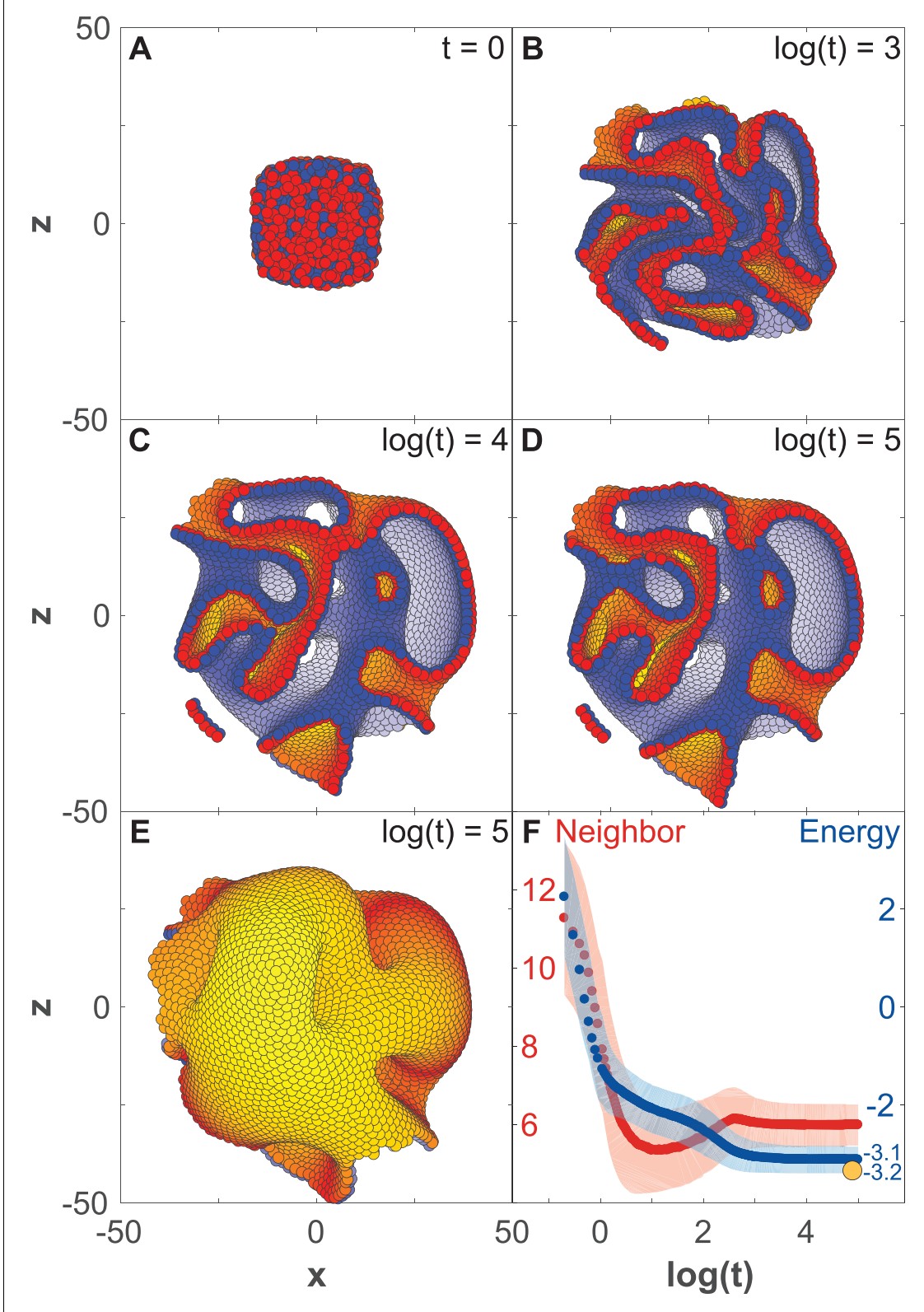

**Figure 3.** Development of 8000 cells from a compact aggregate starting at time 0. (A) Cells are assigned random apical-basal polarity directions and attract each other through polar interactions (see *Equation 6*). (A–D) Cross-section of the system at different time points with red and blue marking two opposite sides of the polar cells. Cells closest to the viewer are marked red/blue, whereas cells furthest away are yellow/white. (E) Full system at the time point shown in (D). (F) Development of the number of neighbors per cell (red) and the energy per cell (blue), as defined by the potential between

*Figure 3 continued on next page*

*Figure 3 continued*

neighbor cells in *Figure 2*. Dark colors show the mean over all cells while light-shaded regions show the cell–cell variations. The yellow dot marks the energy for a hollow sphere with the same number of cells. See *Figure 3—video 1* for full time series. In *Figure 2—figure supplement 1E–G* and *Figure 3—figure supplement 1*, we study how the final morphology depends on noise. In *Figure 3—figure supplement 2*, we show how the outer surface self-seals, and that the shape is maintained when cells divide.

DOI: https://doi.org/10.7554/eLife.38407.010

The following video and figure supplements are available for figure 3:

**Figure supplement 1.** The final shapes are more sensitive to initial polarities than to noise.

DOI: https://doi.org/10.7554/eLife.38407.011

**Figure supplement 2.** The complex morphology in *Figure 3* self-seals and is robust to overall system growth.

DOI: https://doi.org/10.7554/eLife.38407.012

**Figure 3—video 1.** An aggregate of 8000 cells with initial random polarities unfolds into a stable complex morphology.

DOI: https://doi.org/10.7554/eLife.38407.013

## The final shapes are robust to noise but sensitive to initial and boundary conditions

The orientation of polarities in a subpopulation of cells may be set by the environment that the cells are embedded in, for example signaling molecules deposited into extracellular matrix can influence orientation of the AB polarity (*Overeem et al., 2015*) or signals from neighboring cells of a different type can orient PCP (*Chu and Sokol, 2016*). We will refer to these constraints as boundary conditions.

To investigate sensitivity to boundary conditions, we consider three cases where polarities are fixed at all times and point either radially out from the center of mass (*Figure 4A*), radially out from a central axis (*Figure 4B*), or pointing away from a central plane (*Figure 4C*). As anticipated, the difference in symmetries of boundary conditions results in a sphere, a cylinder, or two parallel planes (*Figure 4—video 1*). At the same time, in these symmetric cases, the differences in initial conditions but without imposed boundary conditions are not sufficient to generate different structures; they all converge to the nested 'Russian doll'-like hollow spheres (*Figure 4D*). In development, this highlights the importance of the neighboring tissues for defining boundary conditions.

Our results thus support the idea that polar adhesion enables stable and robust macroscopic shapes. The closest biological parallels would be the complex luminal morphologies emerging in reaggregation experiments on for example *Hydra* (*Seybold et al., 2016*) or in vitro culture of pur-junkie brain cells (*Muguruma et al., 2015*). Together with our simulations, these experiments highlight how stable and complex morphologies can develop in non-proliferating populations from cell rearrangements alone.

## Folding by pressure or rapid proliferation result in different fold-morphologies

Transitions from spheres to folded shapes are ubiquitous in development. Folds are an important part of in vivo organ development, and the composition of cell types in the folded organoids is closer to that in real organs (*Greggio et al., 2013*). To date, it is unclear what drives the transition from spheres to folded lumens. One possibility is that it is driven by the mechanical properties of the matrigel that effectively may place the growing organoid under pressure. Alternatively, data from 3D brain organoids suggests that the rapid cell proliferation leads to the emergence of surface folding (*Li et al., 2017*).

The simplicity of our tool allows to explore both of these scenarios. To model dividing cells, we pick a random cell from the entire population and introduce a new daughter cell with inherited polarity direction placed in a random location a half cell radius away from the mother cell. This event introduces dynamic perturbation by locally increasing cell density and requires some time to relax back to equilibrium. If proliferation is slow, and the time between two cell divisions anywhere in the system is longer than relaxation time of the whole system (the time it takes to reach equilibrium), the system approaches global equilibrium and will expand as a sphere. However, if proliferation is increased, the system will be pushed out of equilibrium and folds will emerge (*Figure 5A*, Materials and methods). In more quantitative terms, our forces are such that a single cell can move up to 0.2

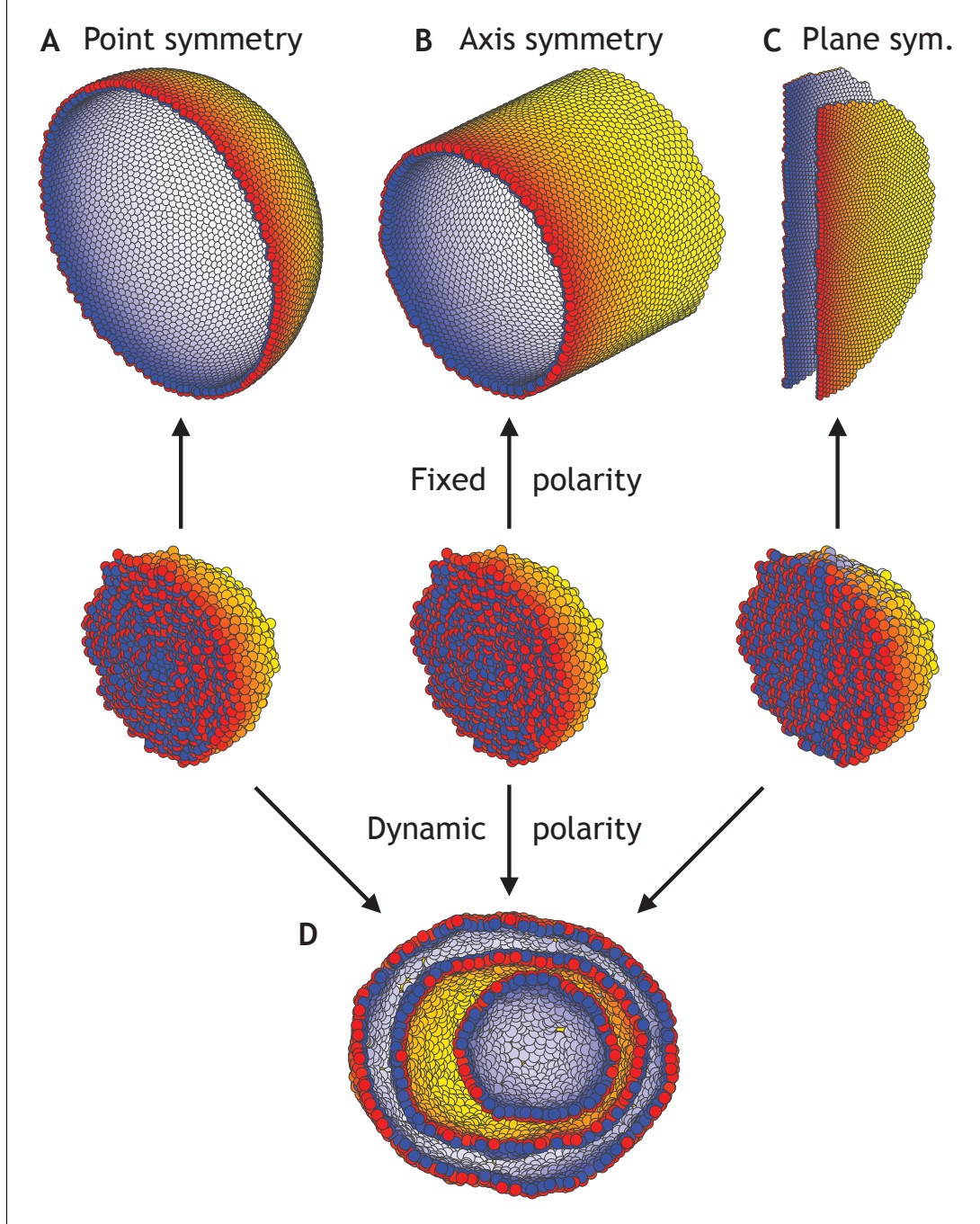

**Figure 4.** Different morphologies can be obtained by varying boundary conditions (*Figure 4—video 1*). (**A**) A hollow sphere emerges if polarities are fixed and initially point radially out from the center of mass. (**B**) A hollow tube is obtained if polarities point radially out from a central axis. (**C**) Two flat planes pointing in opposite directions are obtained if polarities point away from a central plane. (**D**) For all three initial conditions (**A–C**), if the polarities are allowed to change dynamically and the noise is high ($\eta = 10^0$ compared to $\eta = 10^{-1}$ in A–C), the resulting shape consists of three nested 'Russian doll'-like hollow spheres that will never merge due to opposing polarities. In contrast to the random initial condition in *Figure 3*, the initial conditions in (**D**) are symmetric.

DOI: https://doi.org/10.7554/eLife.38407.014

The following video is available for figure 4:

**Figure 4—video 1.** Dynamics when the polarities have restricted orientations.

DOI: https://doi.org/10.7554/eLife.38407.015

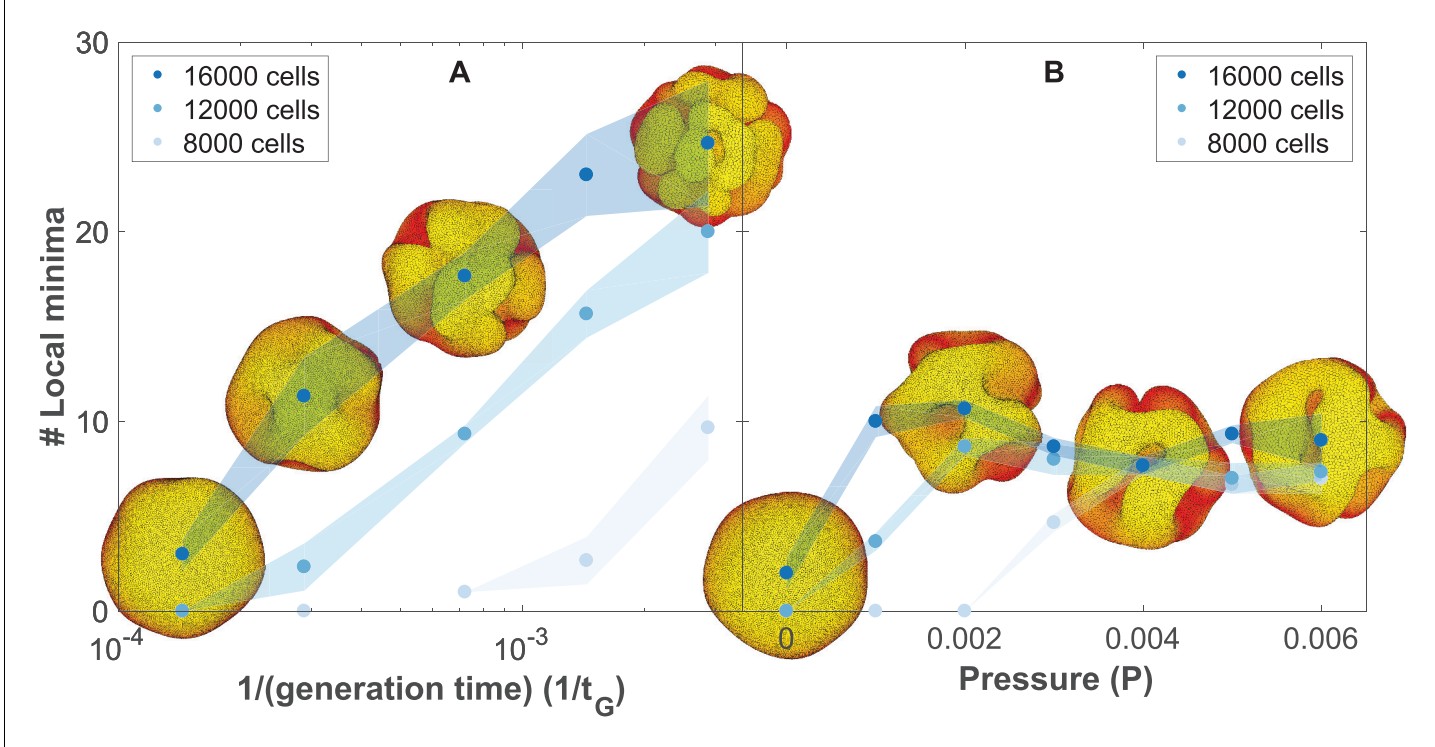

**Figure 5.** The number of complex folds in a growing organoid depends on the generation time and the pressure from the surrounding medium (**Figure 5—video 1**). (**A**) Number of local minima as a function of 1/(generation time), $t_G^{-1}$. In silico organoids grow from 200 cells up to 8000, 12,000, or 16,000 cells with different generation times and no outer pressure. (**B**) Number of local minima as a function of pressure, $P$. In silico organoids grow to the same size with the same 1/(generation time), $t_G^{-1} = 1.4 \cdot 10^{-4}$ but different outer pressure. The images illustrate the 16,000 cells stage. Blue dots mark the average, while light shaded regions show the SEM based on triplicates. See also **Figure 5—figure supplement 1** for additional measurements on the differences between rapid growth and pressure.

DOI: https://doi.org/10.7554/eLife.38407.016

The following video and figure supplement are available for figure 5:

**Figure supplement 1.** Organoids grown under external pressure have deeper and longer folds compared to organoids grown with rapid cell proliferation.

DOI: https://doi.org/10.7554/eLife.38407.017

**Figure 5—video 1.** In silico organoids grown from 200 up to 16,000 cells.

DOI: https://doi.org/10.7554/eLife.38407.018

cell diameter per time unit. For cells that divide every 1000 time units, the transition to non-equilibrium buckling happens when the system has grown to about 5000 cells (**Figure 5—video 1**).

As cells divide faster, our simulations predict a transition from a smooth spherical shell to an increasingly folded structure with multiple pronounced folds, in line with the observation of brain organoids proliferating at different rates (**Li et al., 2017**). In comparison with the model for cortical convolutions by **Tallinen et al., 2016** in which folding is a result of expanding cortical sheet adhered to the non-expanding white matter core, our mechanism does not require a bulk core. Instead folds emerge in a fast-expanding sheet when the growth is faster than the global relaxation to dynamical equilibrium.

While we find that the external pressure is not necessary for folding, pressure alone can also drive folding (**Figure 5B**, Materials and methods). However, this scenario contradicts the observation that pancreatic organoids can grow as spheres or folded morphologies in gels with the same stiffness but different media composition (**Greggio et al., 2013**).

In principle, both scenarios may contribute to folding, but visually the fold morphologies are different. To differentiate between the two, we have quantified the final folded structures in terms of their local minima (**Figure 5**, Materials and methods, see also **Figure 5—video 1**). Our simulations predict that in the pressure-driven case, the number of local minima will reach an upper limit as

organoids increase in size (*Figure 5B*). In the case of out-of-equilibrium proliferation, new folds can continue forming as organoids grow (*Figure 5A*). Increased proliferation causes more and shallower folds. These folds are different than obtained with pressure which causes fewer but deeper minima. Quantitatively, both the depth and the horizontal extension of the folds are about double as large with pressure than with growth-induced folding (*Figure 5—figure supplement 1*).

## PCP enables convergent extension and robust tubulogenesis

Despite the numerous evidences supporting the role of PCP in tubulogenesis, it remains unresolved whether oriented cell division or the extent of CE controls tube length and width (*Karner et al., 2009*; *Carroll and Yu, 2012*). It is also debated if it is important for the tubes to maintain regular shape, or if it is only important for tube initiation and growth (*Kunimoto et al., 2017*).

The simplicity of our approach allows us to address these questions by introducing cell–cell interactions through PCP. This term favors front-rear cell alignment in the interaction potential with only two additional parameters: the strength of the orientational constraint of AB polarity with respect to PCP, $\lambda_2$, and the strength of PCP, $\lambda_3$ (see *Equations 4–8*). For simplicity, we focus on the stability (ability to maintain regular diameter over time) and tube morphogenesis in systems without cell division.

Inducing PCP in a spherical lumen leads to two significant events. First, independent of initial orientation, after some transient time PCPs becomes globally ordered and point in direction parallel to an emerging equator that self-organizes around the sphere (inset in *Figure 6*). This arrangement has the lowest energy. Second, cells start intercalating along the axis perpendicular to PCP orientation, gradually elongating the lumen (*Figure 6—video 1*). During intercalations, cells exchange their neighbors through T1-like transitions as reported experimentally, *Figure 6—figure supplement 3* (*Nishimura et al., 2012*; *Sanchez-Corrales et al., 2018*). The intercalations along the axis continue until the force balance between AB polarity and PCP is restored at a new equilibrium. Thus, our model predicts that the strength of PCP ($\lambda_3$) relative to AB polarity ($\lambda_1$) determines the width and the length of the tube (*Figure 6*). We obtain similar results if we constrain PCP to always remain in the apical plane and thus does not allow PCP to reorient AB polarity (*Figure 6—figure supplement 1*, Materials and methods). Note, that this result is very different in nature from the tube presented in *Figure 4B* as both AB polarity and PCP can now reorient in each cell at any time.

These results support the observations that stable tubes can emerge without cell proliferation. In addition, when first the tube is formed, loss of PCP does not lead to cyst formation as recently shown by *Kunimoto et al. (2017)*. However, localized cysts could result if the lumen is initialized with varying strength of PCP along the axis (*Figure 6—figure supplement 2*).

## Two polarities are sufficient to explain major features of sea urchin gastrulation

Currently invagination in neurulation and gastrulation is understood and quantitatively modeled as a process driven by changes in cell shapes or the mechanical properties of cells with AB polarity (*Rauzi et al., 2013*; *Tamulonis et al., 2011*; *Misra et al., 2016*; *Hočevar Brezavšček et al., 2012*). This process is often assumed to be driven by apical constriction and decoupled from the eventual tube formation and elongation. However, emerging data suggests that PCP drives both invagination and tube elongation (*Nishimura et al., 2012*; *Croce et al., 2006*; *Long et al., 2015*) likely because apical constriction is controlled by PCP (*Ossipova et al., 2014*; *Nishimura et al., 2012*; *Croce et al., 2006*) and cell intercalations, similar to those in CE, contribute to invagination (*Nishimura et al., 2012*; *Rembold et al., 2006*; *Sanchez-Corrales et al., 2018*).

To probe the limits of our approach, we investigated if AB polarity, PCP, and boundary conditions reminiscent of posterior organizer (*Loh et al., 2016*) are sufficient to recapitulate the main stages of sea urchin gastrulation: invagination, tube formation, elongation (by CE), and finally merging of the gastrula tube with the pole opposite to invagination site.

The current understanding is that sea urchin gastrulation consists of primary invagination, driven by swelling of the inner layer of extracellular matrix beneath invaginating cells (*Lane et al., 1993*), and formation of a ring of bottle cells due to apical constriction (*Kimberly and Hardin, 1998*), and secondary invagination where tube elongates due to CE (*Lyons et al., 2012*). PCP is necessary for

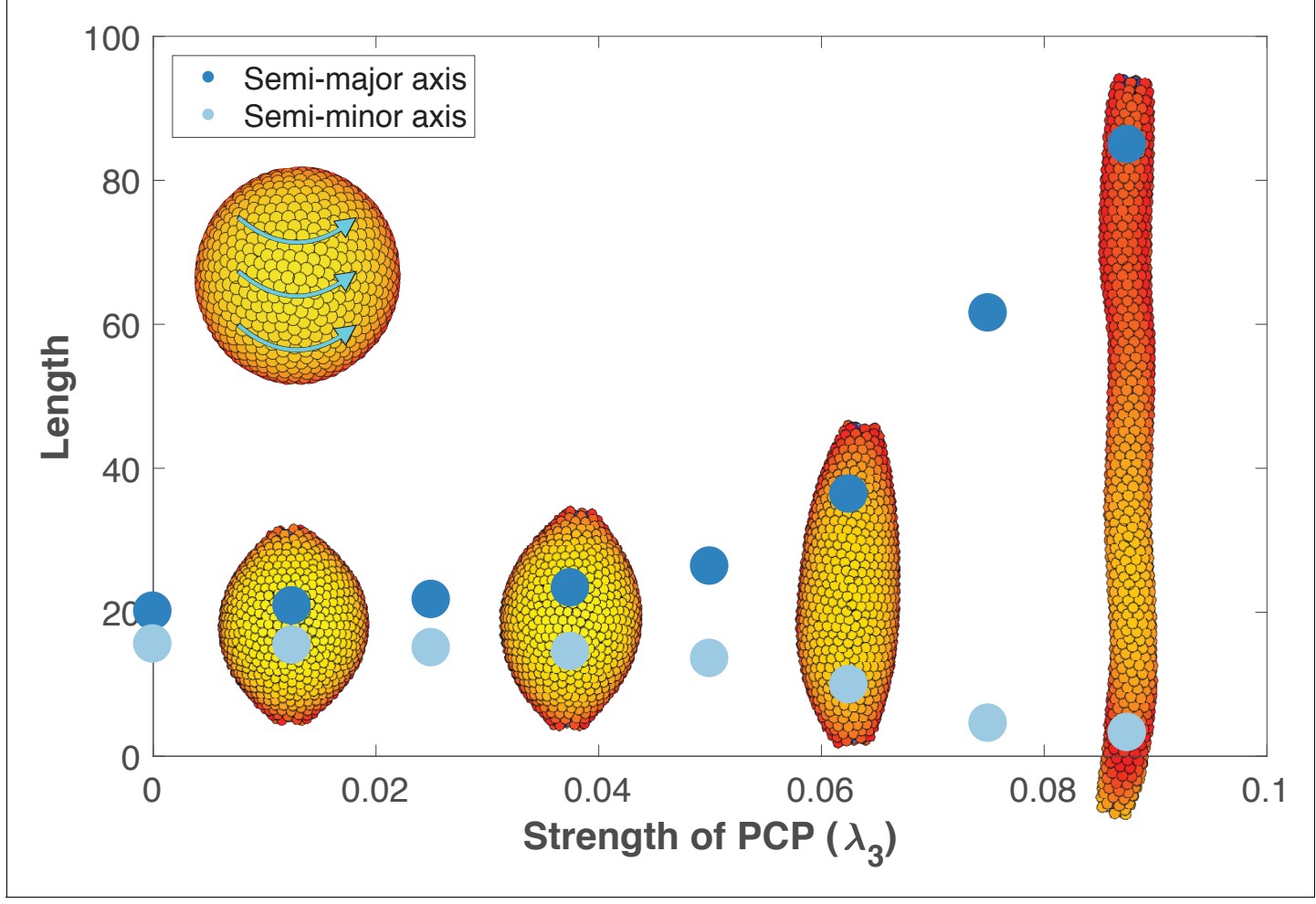

**Figure 6.** The length and width of tubes are set by the strength of planar cell polarity (PCP, $\lambda_3$). For each value of $\lambda_3$, we initialize 1000 cells on a hollow sphere with PCP whirling around an internal axis (PCP orientation marked by cyan arrows in the top-left inset). Semi-major axis (dark blue) and semi-minor axis (light blue) are measured at the final stage (Materials and methods). Images show the final state. Throughout the figure, $\lambda_2 = 0.5$ and $\lambda_1 = 1 - \lambda_2 - \lambda_3$. The animated evolution from sphere to tube is shown in *Figure 6—video 1*. See also *Figure 6—figure supplement 1* where we show that tubes also form when we disable the direct influence of PCP on apical-basal polarity, and *Figure 6—figure supplement 2* where we vary the degree of PCP along the axis of the tube. In *Figure 6—figure supplement 3*, we show that cell intercalations result in experimentally reported T1 neighbor exchanges during convergent extension.

DOI: https://doi.org/10.7554/eLife.38407.019

The following video and figure supplements are available for figure 6:

**Figure supplement 1.** Removing the influence of planar cell polarity (PCP) on apical-basal (AB) polarity.

DOI: https://doi.org/10.7554/eLife.38407.020

**Figure supplement 2.** A lumen forms inside a developing tube in areas that lack planar cell polarity (PCP).

DOI: https://doi.org/10.7554/eLife.38407.021

**Figure supplement 3.** T1 exchanges occur during sphere–tube transition.

DOI: https://doi.org/10.7554/eLife.38407.022

**Figure 6—video 1.** Model of tubulogenesis.

DOI: https://doi.org/10.7554/eLife.38407.023

both invagination, possibly through its effect on apical constriction in bottle cells (*Nishimura et al., 2012*), and tube extension (*Croce et al., 2006*).

Motivated by these observations, we set boundary condition such that PCP of the invaginating cells are oriented around the anterior-posterior (top-bottom) axis, and are always in the apical plane. This constrain on PCP orientation allows for CE. While this particular configuration is not documented, it is consistent with observed effects of WNT orienting PCP within the apical plane

(*Humphries and Mlodzik, 2018*). Second, we simulate the combined effect of bottle cells (*Figure 2—figure supplement 2*) and bending by swelling of the extracellular matrix by applying an external force, *F*, on AB polarity (see Materials and methods). This force gradually reorients AB polarity away from the anterior-posterior axis, thus leading to bending of the epithelial sheet. The effect is maximal for cells closest to the anterior-posterior axis. This external force is a phenomenological description aiming at capturing the observed effects of how change in AB polarity results in tissue bending and does not aim at capturing mechanisms driving the reorientation of AB polarity.

As a result, cells start to rearrange, the bottom flattens (*Figure 7B*) and bends inward (*Figure 7C*). Subsequently, the CE-driven by PCP causes the invaginated cells to rearrange, tube elongates, and merges with the top of the sphere (*Figure 7F* and *Figure 7—video 1*). In line with experimental observations, the tube elongates due to cells moving in the tube (*Martik and McClay, 2017*).

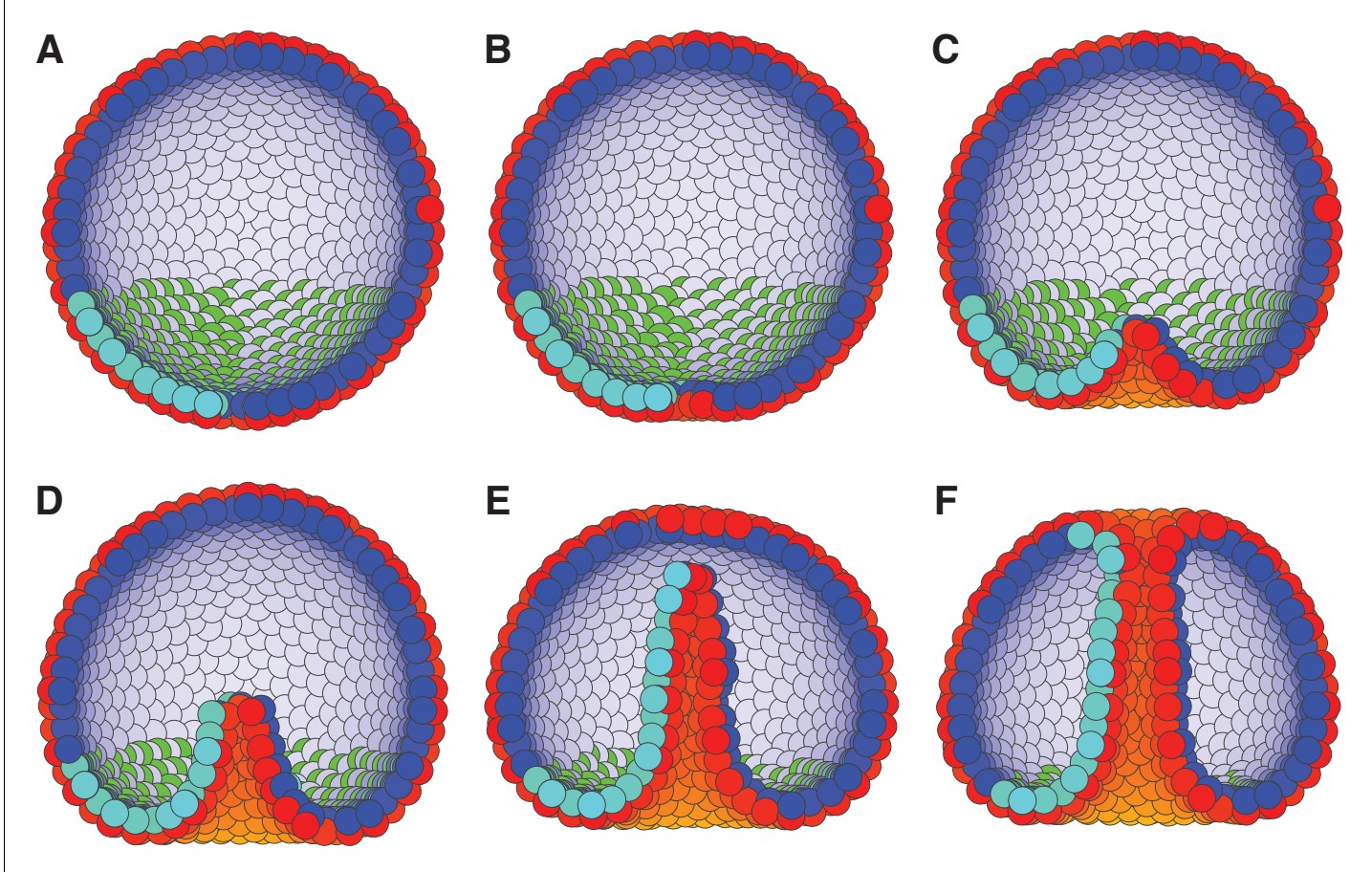

**Figure 7.** External constraints on apical-basal (AB) polarity and planar cell polarity (PCP) can initiate invagination and drive gastrulation in sea urchin. (A) The lower third of the cells in a blastula with AB polarity (apical is blue–white, basal is red–orange) pointing radially out acquire PCP (cyan–green) in apical plane pointing around the anterior-posterior (top-bottom) axis (as in the inset to *Figure 6*). (B) Flattening of the blastula and (C) invagination occur due to external force reorienting AB polarity (Materials and methods). (D–E) Tube elongation is due to PCP-driven convergent extension and (F) merging with the top of the blastula happens when the tube approaches the top. Throughout the simulation, $\lambda_1 = 0.5$, $\lambda_2 = 0.4$, and $\lambda_3 = 0.1$ for the lower cells while the top cells have $\lambda_1 = 1$ and $\lambda_2 = \lambda_3 = 0$. For full time dynamics see *Figure 7—video 1*. In *Figure 7—figure supplement 1*, we consider alternative scenarios of sea urchin gastrulation and and neurulation.

DOI: https://doi.org/10.7554/eLife.38407.024

The following video and figure supplement are available for figure 7:

**Figure supplement 1.** Directed changes in the direction of planar cell polarity (PCP) may drive invagination in gastrulation and neurulation.

DOI: https://doi.org/10.7554/eLife.38407.025

**Figure 7—video 1.** Model of sea urchin gastrulation.

DOI: https://doi.org/10.7554/eLife.38407.026

The contribution of extracellular matrix swelling is specific to sea urchin invagination. As bottle cells alone are sufficient to drive invagination in other systems, we have tested a cell-intrinsic scenario where AB polarities of bottle cell neighbors prefer to be tilted towards each other (by e.g. modifying the potential in *Equation 6* to capture AB polarities as in *Figure 2—figure supplement 2*, unpublished results). This also resulted in successful invagination.

Several observations suggest that radial intercalation movements towards the center of invagination may drive tissue bending (*Sanchez-Corrales et al., 2018*; *Panousopoulou and Green, 2016*; *Rembold et al., 2006*; *Chung et al., 2017*). When tested in our model, PCP alone (omitting external or cell-intrinsic reorientation of AB polarity) results in exvagination and tube elongating outside of the sphere (this is more energetically favorable than extending inwards). A similar, exogastrulating, phenotype was observed in a PCP mutant (*Long et al., 2015*). While the similarity may be accidental, it is possible that this mutation abrogates PCP-driven apical constriction (*Ossipova et al., 2014*; *Nishimura et al., 2012*) and since cellular apices face outside, abrogating apical constriction eliminates the bias in direction of tube formation. We have also tested the consequences of apical constriction (reorienting AB polarity) in the model without PCP and while this was sufficient for invagination, the cavity remained spherically symmetric and failed to form a tube.

## Discussion

Despite the stunning diversity and complexity of morphologies, the same concepts seem to emerge across organismal development. One of them is the link between local, cellular, and global, organs/ whole organism, symmetry breaking. We know, from experimental and, to a lesser degree, theoretical work that cellular polarity is essential for forming axis, complex folded sheets, and interconnected tubes (*Figure 1—figure supplement 1*). What we do not know is why are these shapes so stable, and where do the differences between species and organs come from. To understand the differences, we typically compare genes or gene regulatory networks, thus limiting our understanding to analog processes within a few related species.

### Phenomenological description bridges cell polarity to macroscopic morphologies

To address the origins of morphological diversity and stability across species and organs, we focused on a phenomenological description of polarized cell–cell interactions. This allowed us to bridge local single-cell symmetry breaking events to global changes in morphologies spanning tens of thousands of interacting cells. With this tool at hand, we find that with only a few parameters, we can recapitulate the two global symmetry breaking events: formation of epithelial sheets and folds by cells with AB polarity, and emergence of global axial symmetry (tubes) among cells with PCP.

Remarkably, our results show that interactions among AB polarized cells lead to stable morphologies, that after initial relaxation remain indefinitely in their final configuration. The morphologies are robust to noise, growth, and local damage (*Figure 3—figure supplement 2*). These results may explain how organs and embryos preserve their architecture while growing. Polar cell–cell interactions not only provide clue to the morphological stability, but also point to a simple explanation to the origin of the diversity. We find that the exact morphological details are defined by initial conditions, for example initial positions and orientation of polarities, and boundary conditions, for example polarities restricted to certain direction for a fraction of the cells. It is thus tempting to speculate that diverse shapes do not require multiple interacting morphogen gradients, but can be a result of differences in initial and/or boundary conditions: as for example presence of yolk cells at start and boundary constraints by vitelline membrane (*Schierenberg and Junkersdorf, 1992*; *Wu et al., 2010*).

The diversity of shapes and forms is further enriched by a second symmetry breaking event, PCP, oriented perpendicular to AB polarity. Within our phenomenological framework, addition of PCP component is simple, and requires only two additional parameters: one favoring perpendicular orientation of AB polarity and PCP within a cell, and another, favoring parallel PCP alignments between neighbor cells. These constraints are the coarse-grained representation of the well-established experimental and computational results on intracellular symmetry breaking events and global ordering of planar polarities mediated by cell–cell coupling (*Le Garrec et al., 2006*; *Amonlirdviman et al., 2005*; *Wang et al., 2006*). The first constraint allowed formation of axial

symmetry and in combination with AB polarity, stable tubes, with length and diameter remaining constant with time. The second constraint resulted in cell rearrangements and intercalations consistent with the cell-autonomous CE typically associated with PCP. The patterns of neighbor exchanges during cell-rearrangements are in line with the ubiquitous T1 exchanges through formation and resolution of four cell vertex, *Figure 6—figure supplement 3* (*Nishimura et al., 2012*; *Sanchez-Corrales et al., 2018*). The mechanism of the CE in our model is in line with the results by 'filopodia tension model' where elongated structures of many cells emerge from local cell–cell interactions in a direction defined by PCP (*Belmonte et al., 2016*). The presented formulations of our model captures only some of the known events contributing to CE and does not include PCP-driven changes in cell shape mediated by for example apical constriction (*Nishimura et al., 2012*) or contribution of external forces (*Lye and Sanson, 2011*).

Combining AB polarization and a local induction of PCP in a subpopulation of cells was sufficient to obtain main stages of sea urchin gastrulation: invagination, tube formation, and elongation through CE as well as merging of the tube with the animal pole at the top of the blastula. It is important to notice that the model for gastrulation uses an external force (*Equation 25*) acting on AB polarity, and the model for neurulation uses an imposed external constraint on PCP. This is not fully satisfying, and suggests an extension of the model to capture tissue bending induced by changes in cell shapes at the edge of the region that invaginates.

The existing in silico models treat invagination and CE-driven tube elongation as independent processes (*Figure 1—figure supplement 1*). Recent data, however, suggests that multiple mechanisms (intracellular apical constriction, intercellular directed cell division and cell intercalations, and supracellular actomyosin cables) act simultaneously and contribute to both invagination and tubulogenesis (*Chung et al., 2017*; *Nishimura et al., 2012*; *Ossipova et al., 2014*). Within our approach apical constriction (modeled as reorientation of AB polarity) and CE can act in parallel (*Figure 7*). It will be interesting to parallel recent experimental work (*Chung et al., 2017*) and computationally investigate how a combination of intra-, inter- and supracellular mechanisms contribute to the robustness of tubulogenesis, and to what extent the model can capture the range of observed phenotypes.

It has been proposed that the above mechanisms may all be coordinated by PCP (*Nishimura et al., 2012*). Besides the reported molecular links, a simple logic suggests that these mechanisms cannot be isotropic as in this case the initial bending will result in spherical structures. Thus, apical constriction, cell intercalations, and actomyosin cables have to be anisotropic (planar polarized) in directions consistent with the eventual tube orientation. This anisotropy is reported for both 'wrapping' tubes forming parallel to the epithelial plane, for example neurulation (*Nishimura et al., 2012*), and 'budding' tubes forming orthogonally to the epithelial plane, for example salivary glands (*Chung et al., 2017*; *Sanchez-Corrales et al., 2018*).

As organizing signals such as WNT can induce and orient PCP (*Chu and Sokol, 2016*) within the apical plane, we asked if it is in principle possible to design PCP constraints (not limited to the apical plane) that would result in 'wrapping' and 'budding'. First, pointing PCP out of the plane was sufficient for a sheet of cells to bend (*Figure 7—figure supplement 1*). This is because in our formulation, PCP can drive reorientation of AB polarity and that in its turn is able to bend the sheet (*Figure 7*). Both 'budding' and 'wrapping' were qualitatively captured by the model when the axial and radial anisotropy were set by constraining orientation of PCP for cells within a circle ('budding' in sea urchin) and two stripes of cells (mimicking hinge points in neurulation 'wrapping'). While CE was needed for proper tube forming in sea urchin example, the tube formed without CE in neurulation (*Figure 7—figure supplement 1*).

Thus, the simulations suggest that 'wrapping' in neurulation and gastrulation in *Drosophila* (*Figure 7—figure supplement 1*) vs. 'budding' in sea urchin and organogenesis (*Andrew and Ewald, 2010*; *Zegers, 2014*) may be outcomes of different constraints imposed on PCP. While it is intriguing to speculate that PCP may be oriented out of epithelial plane directly by organizing signals, this may also be an indirect effect of a sequence of intermediate steps. As the organizing signals not only induce and orient PCP but also drive apical constriction (effectively reorienting AB polarity) the PCP may be gradually oriented out of the original plane of epithelium by the following sequence of events: PCP → apical constriction → tilt in AB polarity → tilt in apical plane → PCP out of original epithelial plane. This less precise but simpler interpretation highlights the fact that PCP may drive many of the alternative mechanisms of tubulogenesis and shifts the focus from the differences in

mechanisms driving tubulogenesis to the differences in boundary conditions – a set of constraints imposed on cell polarities by neighboring tissue (e.g. notochord in neurulation and organizer in gastrulation).

## Testable predictions

In addition to our conceptual findings, we propose three testable predictions. First, we predict that two potential mechanisms behind the emergence of folds in pancreatic organoids – matrigel resistance and rapid, out-of-equilibrium, cell proliferation – will result in distinct morphologies. Our results suggest that in case of rapid proliferation, the growing structure will develop many shallow folds close to the surface which later tend to deepen. In contrast, external pressure causes fewer but deeper and longer folds (*Figure 5—figure supplement 1*). And further, as organoids grow in size, the number of folds will reach an upper limit when under pressure, however, in case of rapid proliferation, the number of folds will keep growing (*Figure 5*). Visual inspection of published morphologies seems to support the out-of-equilibrium growth (*Greggio et al., 2013*; *Li et al., 2017*). To assess if the growth is out-of-equilibrium in 3D organoids, one can quantify the distributions of cell shapes (*Cerruti et al., 2013*). Our model thus predicts that quantitative counting of folds and measurements of the fold depth and length relative to the size of the growing structure may discriminate between the alternative hypothesis. Quantification of the folds can be done in in vitro organoids by either phase or confocal fluorescence microscopy of whole-mount immunostained samples (*Greggio et al., 2013*; *Li et al., 2017*). The fold depth and length can be quantified with the same approach as applied to simulated shapes (Materials and methods) in binarized images of the 3D organoid surfaces. As oganoids cannot be cultured without gel supporting 3D growth, it will be necessary to vary both gel stiffness and generation time to uncouple their respective contribution to the folding. The work by *Little, 2017* shows that in brain organoids generation time can be both slowed down and speeded up by either genetic manipulation or by adding small molecule inhibitors of pathways regulating cell proliferation. Unfortunately, changing stiffness of matrigel also changes its biochemical composition and may affect cell proliferation and differentiation. One will have to turn to synthetic hydrogels, where it is now possible to uncouple mechanical and biochemical clues (*Gjorevski et al., 2016*). To illustrate possible applications of our approach, we have only focused on two out of several possible mechanisms that may contribute to folding. The other likely alternative is that folding may result from differences in biomechanical interactions or generation times characteristic to the different cell types. These alternative scenarios are straightforward to consider in our model and will be an exciting venue to explore when more quantitative data on differences in organoid morphologies is available.

Our second prediction is that in case of tubes formed by non-proliferating cells, the length and width of the tubes are controlled by the relative strength of AB polarity and PCP. This result calls for quantification of adhesion proteins along the AB polarity and PCP axes. In PCP mutants with shorter and wider tubes, one would expect less planar polarization in adherens junctions and actomyosin, for example larger spread compared to wild type in their orientation quantified relative to the tube axis (*Nishimura et al., 2012*). Alternatively, the balance between AB polarity and PCP can be altered by weakening AB polarity, for example mutating tight junction proteins should result in longer tubes. A similar phenotype has already been reported for *Drosophila* tracheal tube (*Laprise et al., 2010*). With the recent advancements in in vitro systems of tubulogenesis, allowing for easy genetic manipulations and more amenable for quantitative imaging, it may in principle be possible to relate the extent of planar anisotropy in PCP mutants and strength of AB adhesion in tight junction mutants with tube length and diameter. The existing coupling between PCP and AB polarity may, however, make it challenging to tweak one polarity at a time.

Our third, and probably most challenging to test, prediction is on the conditions differentiating between tubes forming perpendicular (e.g. sea urchin gastrulation) or parallel (as in *Drosophila* gastrulation or neurulation) to the plane of epithelium. We predict that the outcome will be defined by the orientation of PCP in the invaginating region and the geometry of the boundaries (circular for budding and axial for wrapping) set by for example WNT organizing signals. Recent development in imaging localization of PCP complexes in single cells (*Wu et al., 2013*; *Chu and Sokol, 2016*; *Minegishi et al., 2017*; *Habib et al., 2013*) allows monitoring localization of PCP complexes, and thus PCP orientation, in individual cells. By placing WNT-soaked (*Habib et al., 2013*) beads or WNT-secreting cells (*Chu and Sokol, 2016*) one can vary PCP orientation in the cells at the epithelial

boundary facing WNT and test for the direction of the epithelial bending and possibly tube formation.

Our approach is by purpose phenomenological and by its nature cannot make predictions about specific molecular details. In all cases, we do not see our simulations as finalized predictions, but rather as pointing in the most promising direction for further exploration of these complex developmental processes. Our setup easily allows for changes as we learn more. The proposed tool should be used in close collaboration with gained experimental knowledge on initial conditions, cell generation times and differentiation processes where polarities play a central role.

Our results open for a series of biological generalizations both in development and diseases. On one hand, we now may be able to explain and unify the apparently very distinct morphological transitions during gastrulation in flies, frogs, fish, mice, and humans by accounting for different initial and boundary conditions. Our model suggests how a moderate change in expression of polarities during some critical evolutionary stages could lead to widely different final morphologies. Thereby, development driven by cell–cell polarity interactions could provide major morphological transitions from local and transient modulations in polarity.

On the other hand, it becomes possible to think of gastrulation, neurulation, tubulogenesis, and organogenesis as the same class of phenomena, where the orientation of the tube is guided by local organizers, and lengths/widths of the tubes are determined by the relative strength of AB polarity and PCP. At the same time, there is an emerging view that wound healing and cancer are local perturbations – for example local loss of cells, dysregulation of cell polarities (*Martin-Belmonte and Perez-Moreno, 2012*), proliferation, or autonomously induced organizing signals – of otherwise conceptually the same developmental processes (*Humphries and Mlodzik, 2018*). The power of our model is that it allows to address these hypotheses through predictive models for the dynamics of many cells that interact through combinations of AB polarity and PCP.

## Materials and methods

Throughout the paper, we use the Euler method to integrate the ordinary differential equations stated in the Model section with dt set to 0.1 or 0.2. Lower dt values will give qualitatively similar results but with increased simulation time. Higher dt values, will result in a collapse of the presented morphologies. The time unit is arbitrary, and the same throughout the paper.

The noise parameter $\eta = 10^{-4}$ where nothing else is stated. Lower noise values will give more smooth simulations, while $\eta$ on the order of $10^0$ will result in collapsing shapes (see also *Figure 2—figure supplement 1E–F*).

### Model details

In our model, we use the following potential to describe the pairwise interaction between cells

$$V_{ij} = e^{-r_{ij}} - S\, e^{-r_{ij}/\beta}, \tag{12}$$

where $r_{ij}$ is the center–center distance between cell $i$ and cell $j$, and $S$ is the polarity factor

$$S = \lambda_1 S_1 + \lambda_2 S_2 + \lambda_3 S_3 \tag{13}$$

Here, $\lambda_1$, $\lambda_2$, and $\lambda_3$ are the strengths of the respective polarity terms which are given as

$$S_1 = \left(\hat{p}_i \times \hat{r}_{ij}\right) \cdot \left(\hat{p}_j \times \hat{r}_{ij}\right), \tag{14}$$

$$S_2 = \left(\hat{p}_i \times \hat{q}_i\right) \cdot \left(\hat{p}_j \times \hat{q}_j\right), \tag{15}$$

$$S_3 = \left(\hat{q}_i \times \hat{r}_{ij}\right) \cdot \left(\hat{q}_j \times \hat{r}_{ij}\right). \tag{16}$$

The unit vectors $\hat{p}_i, \hat{p}_j$ and $\hat{q}_i, \hat{q}_j$ represent the apical-basal polarity and planar cell polarity of cell $i$ and $j$. Throughout the paper, $\beta$ is a constant which we set to 5. In order to use the Euler method, we need the gradient of $V_{ij}$ differentiated with respect to position, $\bar{r}_i$, and the two polarities, $\bar{p}_i$ and $\bar{q}_i$:

$$\frac{dV_{ij}}{d\bar{r}_i} = e^{-r_{ij}/\beta}\left\{ \gamma\,\hat{r}_{ij} - \frac{\lambda_1}{r_{ij}}\left[(\hat{r}_{ij}\cdot\hat{p}_j)\hat{p}_i + (\hat{r}_{ij}\cdot\hat{p}_i)\hat{p}_j\right] - \frac{\lambda_3}{r_{ij}}\left[(\hat{r}_{ij}\cdot\hat{q}_j)\hat{q}_i + (\hat{r}_{ij}\cdot\hat{q}_i)\hat{q}_j\right] \right\}, \tag{17}$$

$$\frac{dV_{ij}}{d\bar{p}_i} = e^{-r_{ij}/\beta}\left\{ \lambda_1\left[S_1\hat{p}_i - \hat{p}_j + (\hat{r}_{ij}\cdot\hat{p}_j)\hat{r}_{ij}\right] + \lambda_2\left[S_2\hat{p}_i - (\hat{q}_i\cdot\hat{q}_j)\hat{p}_j + (\hat{q}_i\cdot\hat{p}_j)\hat{q}_j\right] \right\}, \tag{18}$$

$$\frac{dV_{ij}}{d\bar{q}_i} = e^{-r_{ij}/\beta}\left\{ \lambda_2\left[S_2\hat{q}_i - (\hat{p}_i\cdot\hat{p}_j)\hat{q}_j + (\hat{p}_i\cdot\hat{q}_j)\hat{p}_j\right] + \lambda_3\left[S_3\hat{q}_i - \hat{q}_j + (\hat{r}_{ij}\cdot\hat{q}_j)\hat{r}_{ij}\right] \right\}. \tag{19}$$

In order to derive *Equations 17, 18, and 19*, we have used the following:

$$\frac{d}{d\bar{r}_i}e^{-r_{ij}/\beta} = \frac{1}{\beta}\,e^{-r_{ij}/\beta}\,\hat{r}_{ij}, \tag{20}$$

$$\gamma = e^{-r_{ij}(\beta-1)/\beta} - \frac{S}{\beta} + \frac{2}{r_{ij}}\left[\lambda_1\left(\hat{r}_{ij}\cdot\hat{p}_i\right)\left(\hat{r}_{ij}\cdot\hat{p}_j\right) + \lambda_3\left(\hat{r}_{ij}\cdot\hat{q}_i\right)\left(\hat{r}_{ij}\cdot\hat{q}_j\right)\right], \tag{21}$$

$$\frac{dS_1}{d\bar{r}_i} = \frac{1}{r_{ij}}\left[(\hat{r}_{ij}\cdot\hat{p}_i)\hat{p}_j + (\hat{r}_{ij}\cdot\hat{p}_j)\hat{p}_i\right] - 2\left[(\hat{r}_{ij}\cdot\hat{p}_i)(\hat{r}_{ij}\cdot\hat{p}_j)\hat{r}_{ij}\right], \tag{22}$$

$$\frac{dS_1}{d\bar{p}_i} = \frac{1}{p_i}\left[\hat{p}_j - (\hat{r}_{ij}\cdot\hat{p}_j)\hat{r}_{ij} - S_1\hat{p}_i\right], \tag{23}$$

where $p_i$ is the length of the polarity of cell $i$ which is equal to one at all times.

## Generation time in growing organoids

In *Figure 5*, the number of cells, $N$, at a given time, $t$, is $N = 200\,\exp(\ln(2)\,t/t_G)$ where $t_G$ is the generation time. In these simulations, the AB potential (*Equation 6*) between cells is set to zero when the angle between their polarity is larger than $\pi/2$.

## Modeling resistance from matrigel

In *Figure 5*, we model resistance from the matrigel by imposing a surface force pointing toward the center of mass. The potential of the pressure in the growth medium is given by

$$V_M = -\frac{Pr^2}{2r_{max}} \tag{24}$$

where $P$ is the stiffness of the medium, $r$ is distance from the center of mass, and $r_{max}$ is the distance to the cell that is the furthest away from the center of mass. The resulting force will be constant in time at the periphery. Thus, all cells on a growing sphere will be exposed to a force of equal size. However, cells that end up deep inside a folded morphology will experience weaker resistance.

## Quantification of the local minima

In *Figure 5*, the number of local minima is defined as the number of cells that do not have any neighbor cells that are closer to the center of mass than themselves, and at the same time have an average angle between their AB polarity and their neighbor cells displacement vector that is less than $\pi/2$.

## Measuring the tube length and width

In *Figure 6*, the semi-minor and semi-major axes correspond to the half-width and half-length of the tubes, respectively. As cells on opposite sides of a tube have AB polarity pointing in opposite directions, we approximate the semi-major and semi-minor axes, by finding the half of the maximum and minimum distance between two cells with AB polarity pointing in opposite directions.

## Modeling cells with different polarities

In the gastrulation simulation (*Figure 7*), each cell is assigned a specific value of polarity strengths ($\lambda_{1,i}$, $\lambda_{2,i}$, and $\lambda_{3,i}$). We define the mutual interaction strength between a pair, $i$ and $j$, of cells in *Equation 4* with different polarity strengths by setting $\lambda_1 = \text{mean}(\lambda_{1,i}, \lambda_{1,j})$, and $\lambda_2 = \text{mean}(\lambda_{2,i}, \lambda_{2,j})$ as well as $\lambda_3 = \text{mean}(\lambda_{3,i}, \lambda_{3,j})$. This choice makes sure that two neighbor cells interact with a force with equal magnitude but opposite sign. Furthermore, it makes sure that $\lambda_1 + \lambda_2 + \lambda_3 = 1$ holds for all cells.

## Disabling PCP's effect on AB polarity

In the model described by *Equations 6–9*, AB polarity and PCP can influence each other's orientation. To constrain PCP to the apical plane and thus disable its influence on AB polarity, we set $\lambda_2 = 0$ when updating AB polarity in time (during the numerical integration of *Equation 10*).

## Modeling bottle cells and apical constriction by an external force

The invagination in gastrulation is implemented by adding an external force, $F$, that act on the AB polarity in addition to our usual intrinsic forces from *Equation 10*

$$F = -k \, r \, e^{\left(-x^2 - y^2\right)/\sigma^2}, \tag{25}$$

The two parameters are $k$ which is the strength of the force (in *Figure 7*, $k = 0.02$), and $\sigma$ which defines the decline of the gaussian force (in *Figure 7*, $\sigma = 10$). $r$ is the unit vector pointing from $x = y = 0$ to the cells' position, and $x$ and $y$ are the respective coordinates of the cells. This force is applied on the AB polarity and bends the orientation of the polarity away from a $z$-axis (the anterior-posterior axis).

## MATLAB script

MATLAB script to generate and visualize data. MATLAB R2016b or newer is required together with the Statistics and Machine Learning Toolbox. In addition, the Parallel Computing Toolbox is required if the *PAR* parameter in the basic script is set to 1. The input folder contains initial conditions for three standardized systems. Bulk systems have neither apical-basal (AB) polarity nor planar cell polarity. Plane and shell systems have only AB polarity. 'N' in the file names gives the number of cells in the system. The initial polarity directions can be modified on line 4–5 in the basic.m file. Inside this file, it is also possible to set the degree of noise ($\eta$), the size of the time steps (dt), and the relation between the polarity strengths ($\lambda_1$, $\lambda_2$, and $\lambda_3$). The parameter *inc* is used to speed up the simulations by only applying the neighborhood function to the nearest 100 neighbors. Generated data is saved in the output folder, and the visualization script is in a separate folder.

## Acknowledgements

We thank Anne Grapin-Botton for insightful discussions on organoids and Sigurd Carlsen for discussion on tube formation. This research has received funding from the Danish National Research Foundation (grant number: DNRF116) and the European Research Council under the European Union's Seventh Framework Programme (FP/2007 2013)/ERC Grant Agreement number 740704.

## Additional information

### Funding

| Funder | Grant reference number | Author |
| --- | --- | --- |
| Danmarks Grundforsknings-fond | DNRF116 | Silas Boye Nissen<br>Ala Trusina |
| Seventh Framework Programme | FP/2007/2013/ERC no. 740704 | Kim Sneppen |

The funders had no role in study design, data collection and interpretation, or the decision to submit the work for publication.

## Author contributions
Silas Boye Nissen, Ala Trusina, Kim Sneppen, Conceptualization, Resources, Data curation, Software, Formal analysis, Supervision, Funding acquisition, Validation, Investigation, Visualization, Methodology, Writing—original draft, Project administration, Writing—review and editing; Steven Rønhild, Software, Writing—review and editing

## Author ORCIDs
Silas Boye Nissen (iD) http://orcid.org/0000-0002-9473-4755
Ala Trusina (iD) http://orcid.org/0000-0003-1945-454X
Kim Sneppen (iD) http://orcid.org/0000-0001-9820-3567

## Decision letter and Author response
Decision letter https://doi.org/10.7554/eLife.38407.032
Author response https://doi.org/10.7554/eLife.38407.033

# Additional files

## Supplementary files
• Source code 1. MATLAB script.
DOI: https://doi.org/10.7554/eLife.38407.027

• Transparent reporting form
DOI: https://doi.org/10.7554/eLife.38407.028

## Data availability
All data generated or analyzed during this study are included in the manuscript and supporting files. MATLAB code to reproduce or generate new data is added as a supplementary zip file together with a MATLAB script to visualize the data.

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
