## [Decision Letter]

Thank you for submitting your article "Theory bridging cell polarities with development of robust complex morphologies" for consideration by *eLife*. Your article has been reviewed by Naama Barkai as the Senior Editor, a Reviewing Editor, and three reviewers. The reviewers have opted to remain anonymous.

The reviewers have discussed the reviews with one another and the Reviewing Editor has drafted this decision to help you prepare a revised submission.

The reviewers and Editor greatly appreciated the novelty of your theory that will hopefully stimulate discussion, experiments and new work. They found the paper a creative, and thought-provoking contribution to the field of development. It puts the emphasis on local and rather generic interactions between cells' polarities. It is a worthwhile goal to illustrate the range of morphogenetic events that could occur based on the assumed cellular behaviors, even if they reflect biological reality to greater or lesser extents. Nevertheless, the reviewers raise a number of concerns, summarised below. Please consider all of the reviewers' points. The reviewers often suggest presentation ideas since they do think the work is worthwhile, even if it is sometimes controversial it offers a good starting point for new models and discussion.

- The reviewers think, if the model is to inform our understanding of morphogenesis, that the mechanical forces in the model reflect, at least loosely, a physically plausible state of forces in the tissue. This is a very serious concern. Specifically, reviewer 1 raises serious concerns about the origins of tissue rotation that absolutely needs to be addressed. reviewer 2, from a very different perspective, seconds these concerns, worrying about the ability of PCP axes to influence AB axes. These points are better explained in the reviewer comments.

- All reviewers raised concerns about the sea urchin example and it would probably be useful to find a good way of either discussing this example. One suggestion is to use the current manuscript to show what is possible. The sea urchin example is (probably – unless we are misunderstanding things) implausible, but is a possible outcome given the input assumptions, and we'd therefore suggest not to eliminate it but discuss the plausibility. In the experts’ opinion, the initial conditions for neurulation are equally implausible, but still demonstrate the possibilities of the model.

- In short, a clear description of assumptions and their influence on the results would be much appreciated.

- In general, as you can easily see some of the reviewers are physicists, and some are biologists. It is a hard exercise but please try and rewrite the manuscript to appeal to both (e.g. define terms that are obvious to physicists, be critical in discussing biological reality for the ignorant physicist), as it is clear it does have the potential to influence both groups of researchers. There is no need to oversell.

- A discussion of testable predictions needs to be improved by making more explicit what manipulations would be performed and how outcomes would be analyzed.

*Reviewer #1:*

This manuscript presents a cell-based model for the morphogenesis of 3D epithelial structures. Cells, described as point particles, are endowed with two-unit vectors representing apico-basal (AB) polarity and planar cell polarity (PCP). Cells interact through a pair potential that is modulated according to the orientation of the polarity vectors. The positions of the cells and the orientations of the polarity vectors follow a simple relaxational dynamics, with a stochastic term.

The introduction of a minimal model to study the collective dynamics of polarized cells, in the context of epithelial morphogenesis, is of potential interest. However, to be of value, such a model should present a physically consistent description of the phenomena of interest. I am unclear that this is the case of the present model. Although its energy function is invariant under simultaneous rotation of the cell positions and polarity vectors, the dynamics as written here (below equation 8) can give rise to a net torque and rotation of the system, which should not be if motion is driven by internal forces. To rephrase my concern, if I understand the model correctly, the orientation-dependence of the energy means that the tissue as whole can rotate to orient in a specific direction, like a compass in a magnetic field (I think that's part of how flat sheets are made in one example). But for a freely-floating tissue there isn't an extrinsic force like the magnetic field to rotate it around, and it's not a motion that can occur through intercellular forces.

This is particularly evident when the polarities are fixed: with a fixed AB polarity alone (as in Figure 4), a doublet of cells minimizing its energy would rotate to align perpendicular to their polarity vectors (cf. the coupling term in equation 6). As far as I understand it, the same effect is at play in the application of the model to sea urchin gastrulation (Figure 7): the imposition of a fixed PCP orientation, through a displacement of AB polarity, gives rise to an extrinsic torque that bends the tissue. If this interpretation of the model is correct, if there's an extrinsic force imposing the orientation of the surface at every point, the model is not a serious contender to explain how an urchin is made. It's maybe less critical for other examples but leaves me unclear how seriously the possibilities illustrated by the model can be taken. That being said, the model certainly stands as a stimulant for further thinking along the same lines.

Similarly, in the case of fold formation on a sphere (Figure 5), which the authors suggest can occur if cell proliferation is on a faster time scale than relaxation, the physical origin of a slow relaxation time scale, and the mechanism of the observed instability, would merit discussion. Although assuming relaxational dynamics is standard and justified when the local minima of the energy are of interest, it might be questioned when turning to dynamical phenomena. Do folds develop through buckling, because the sphere grows (in number of cells) faster than it can expand in the surrounding medium? In many contexts, mechanical equilibration is much faster (seconds or minutes) than proliferation (hours). What would be the origin of a longer time scale for relaxation? Resistance from the surrounding medium might be one, but the authors present fast proliferation as an alternative mechanism to pressure from growth in a stiff gel.

I also have some issues of form with the manuscript (e.g. I do not find the connection that the authors draw between the model and evolution very substantive), but the above questions on the physical relevance of the model are the main issues that I would like to see clarified before the manuscript is given further consideration.

*Reviewer #2:*

This manuscript describes a mathematical model and associated simulations of large clusters of cells that can be assigned properties of apical basal polarity and planar cell polarity and may proliferate, to follow their 3D morphological evolution. The aim is to test the hypothesis that "stability, complexity, and diversity are emergent properties in populations of proliferating polarized cells." The authors conclude that "polarity enables complex shapes robust to noise but sensitive to changes in initial and boundary constraints." The work is of potential interest, but a number of substantial issues should be addressed. Most important, some essential definitions are not provided, the writing needs to be made more accessible to biologists who are not modelers, some key assumptions should be made explicit, and others.

Essential revisions:

Definitions needed: What precisely is meant by "stability?" Are the authors referring to feature constancy over time? Do they mean that models will converge to a specified set of features? What features? For example, they suggest that the trajectory in Figure 3 is stable, but noise seems to qualitatively change the shape (Figure 2—figure supplement 1), though it still produces a complex folded shape. It's not clear what, exactly, the authors are claiming.

I struggled with the idea of "boundary conditions." I eventually concluded that, here, the term appears to mean restrictions on polarity orientations, and maybe other parameters. However, to a biologist, "boundary conditions" means something very different. Difficulty with this passage was compounded by the challenging sentence "At the same time, in these symmetric cases the differences in initial conditions (no boundary conditions) are not sufficient to generate different structures." This implies that differences in initial conditions means no boundary conditions. I think the authors mean to say "differences in initial conditions but without imposed boundary conditions." Finally, in the Discussion section the authors say, "boundary conditions, e.g. fixed polarities for a fraction of the cells in some time." A precise definition should be provided when the idea of boundary conditions is first introduced.

"Minima" is defined in the Materials and methods section, but the authors also use "maxima." By analogy, one can guess what this means, but please define.

Improve writing to make accessible to biologists: I'm a biologist with ample experience collaborating with modelers. Still, I found the manuscript difficult to follow and needed several readings to understand. If the goal is to make this work accessible to a broad audience (and it should be if published in *eLife*), the authors should carefully revise the text to replace jargon with precise simple language.

Describe implications of model structure: Several features of the model are not as explicit as they should be. One is the conflation of PCP with convergent extension behavior. PCP defines a vectorial molecular asymmetry, and PCP can have a number of outcomes. One is CE, but others include the asymmetric elaboration of cellular structures, or the asymmetric induction of gene expression that results in different cell fates, for example. Here, the model structure equates PCP to forces that drive CE. Furthermore, the descriptions of this implementation are misleading. In the Introduction, the authors, in describing the implementation of AB and PCP polarity, say "In other words, the adhesion strength between neighbor cells is modulated by the orientation of their polarities." Are the authors describing how AB and PCP (=CE) are implemented in the model, or how they function biologically? There is no evidence that molecular PCP functions by regulation of adhesion (involvement of Celsr proteins notwithstanding), and differential adhesion is only one part of a complex and only partially understood set of mechanisms that drive CE.

Another implicit part of the model as I understand it, and one that is probably very important to the outcomes, is the notion that altering PCP vectors can influence the AB vectors of cells. I know of no biological precedent to justify this behavior. This input assumption is inconsistent with my understanding of the biological reality. AB polarity is an initial condition that must be met for the subsequent establishment of PCP, which is then by definition orthogonal to the AB axis. There are examples in which PCP determines edges of groups of cells that then increase tension, thereby inducing folds, but this is a different mechanism from the one implied here. I think it would be interesting for the authors to explore how their model would operate if AB polarities orient PCP, but PCP cannot directly influence AB. I suspect that the examples of tube formation would fail. At a minimum, they should discuss the need for this assumption explicitly, noting that it is not currently supported by biological data (or citing the data if there is some that I'm not aware of).

Some unlikely assumptions: While it may be interesting that complex forms such as the sea urchin-like gastrulation and the folded neural tube CAN be produced within the confines of the model, the conditions needed stretch the bounds of plausibility. The model simulating sea urchin gastrulation requires a precise choreography of PCP parameters that could happen but seem unlikely. Even more difficult to believe is the initial conditions invoked to simulate neural plate folding. Specifically, invoking "two rows where the PCP points parallel and antiparallel to AB polarity" invokes conditions for which I know of no precedent at all.

Additional discussion of testable predictions needed: Re-emergence of folds in pancreatic organoids, pressure causes fewer and much deeper folds that form early during growth. This seems to be a qualitative measure. How would biological observations be assigned to one or the other category? Isn't the key difference that the pressure hypothesis predicts that the number of minima maxes out, whereas the proliferation model predicts that the number keeps growing with increase in size (Figure 5)?

Re tubes formed by nonproliferating cells, "in PCP mutants with shorter and wider tubes one would expect higher fraction of "AB adhesion" relative to "PCP adhesion". However, as noted above, PCP is not directly related to any known adhesion events.

Re the conditions differentiating between tubes forming perpendicular (e.g. sea urchin gastrulation) or parallel (as in *Drosophila* gastrulation or neurulation) to the plane of epithelium, the measurements could in principle be made, but as noted above, the conditions seem to me to be highly implausible, and ask us to ignore the likely contribution of other signals.

*Reviewer #3:*

This is a creative and innovative article on development. The authors introduce a simple theoretical model based on the idea that apical-basal (AB) polarity and planar cell polarity (PCP) are (the) main driving forces relevant for the formation of complex tissue morphologies. The model idealises cells as point particles with two independent axes (polarities), an AB and a PCP axis. Pairwise interactions between cells contain an attractive and a repulsive component where the magnitude of the attractive component depends on three additive factors: alignment of the AB axis, alignment between AB and PCP axis, and alignment of PCP axis. Noise comes into the model at various levels: Langevin noise in the equations of motion, noise at the level of choosing the axes orientation at cell proliferation, and noise at the level of initial conditions. In summary, this seems to be a computationally efficient model to explore the question: How can cell-cell adhesion that depends on the relative orientation of polarisation axes alone drive different morphologies during development?

Indeed, the authors simulations show that their model is able to reproduce a broad range of morphologies depending on several factors: choice of initial conditions, relative tendencies of the various axis to orient with respect to each other, and cell proliferation rate. A clear strength of the computational model (I would not call it a theory) is that it provides a mechanistic link between morphology and cell-cell interaction as well as initial conditions. Thus, I find it a clear conceptual advance in the field which puts a fresh perspective on the role of generic interactions for the emergence of different tissue morphologies. The authors also list a few testable predictions. Here, I am not so convinced yet for very basic reasons: Testable predictions require either a clear link between a molecular and an emergent process, or a set of mutually exclusive alternatives that can be tested. Concerning the latter case (in the section of organoids) the authors analyse two possible scenarios (cell proliferation and a specific way to implement external forces). These are not two mutually exclusive alternatives but only two out of many possible conceivable scenarios. Concerning the former case, in the example on sea urchin gastrulation the authors seems to a large degree put in what they would like to get out. Finally, and this is more on the side of a personal opinion, I am not convinced that morphologies that strongly depend on initial conditions are a robust way for cellular development. For instance, how can an organism make sure that all these initial conditions are properly chosen?

In summary, I found the paper a well-written, creative, and thought-provoking contribution to the field of development. It puts the emphasis on local and rather generic interactions between cells' polarities. If the authors can convincingly reply to my above queries, I would be inclined to recommend it for publication.

---

## [Author Response]

Reviewer #1:

This manuscript presents a cell-based model for the morphogenesis of 3D epithelial structures. Cells, described as point particles, are endowed with two-unit vectors representing apico-basal (AB) polarity and planar cell polarity (PCP). Cells interact through a pair potential that is modulated according to the orientation of the polarity vectors. The positions of the cells and the orientations of the polarity vectors follow a simple relaxational dynamics, with a stochastic term.The introduction of a minimal model to study the collective dynamics of polarized cells, in the context of epithelial morphogenesis, is of potential interest. However, to be of value, such a model should present a physically consistent description of the phenomena of interest. I am unclear that this is the case of the present model. Although its energy function is invariant under simultaneous rotation of the cell positions and polarity vectors, the dynamics as written here (below equation 8) can give rise to a net torque and rotation of the system, which should not be if motion is driven by internal forces.

One of the implications of the coupling between position and polarity is that the rotation of the system can indeed result in absence of external forces on cell position, but if external force is applied to cell polarity (e.g morphogen gradients or change in cell shape, illustrated in Figure 2—figure supplement 2). In a sheet of cells, turning AB axis in one cell would result in rotation of its neighbors Figure 2—figure supplement 3 (and Figure 2—video 1). In case of two cells (Figure 2—figure supplement 3C–D and Figure 2—video 1C–D), a somewhat similar physical system would be two spheres connected by a rod. A finite rotation of one sphere (without change of its position) will rotate the rod and thus displace the other sphere.

We have now added this implication to our subsection ”Model implication”.

To rephrase my concern, if I understand the model correctly, the orientation-dependence of the energy means that the tissue as whole can rotate to orient in a specific direction, like a compass in a magnetic field (I think that's part of how flat sheets are made in one example). But for a freely-floating tissue there isn't an extrinsic force like the magnetic field to rotate it around, and it's not a motion that can occur through intercellular forces.

**Author response image 1. respfig1:** Changing the polarized direction of a plane of cells does not rotate the plane as a whole but breaks it into smaller planes. Time *t* = 0 shows a plane consisting of 500 cells with AB polarity pointing to the right. At time *t* = 0.1, the direction of the polarity is shifted by 45 degrees. Since the polarity is fixed in time, the planes break into smaller pieces, and at time 10^4^ they have merged into two planes that are separated by several cell diameters.

To parallel the example with a plane in a magnetic field, we have simulated a plane of 500 cells in which AB polarity is turned by 45 degrees and maintained at that new angle at all times. Here there is no external field on cell positions in our potential but a constant external constraint on the AB direction. Then the subsequent relaxation dynamics will move cells, disrupting the epithelial sheet in many tilted subplanes. Thus the sheet does not rotate as a connected structure.

This is particularly evident when the polarities are fixed: with a fixed AB polarity alone (as in Figure 4), a doublet of cells minimizing its energy would rotate to align perpendicular to their polarity vectors (cf. the coupling term in equation 6). As far as I understand it, the same effect is at play in the application of the model to sea urchin gastrulation (Figure 7): the imposition of a fixed PCP orientation, through a displacement of AB polarity, gives rise to an extrinsic torque that bends the tissue. If this interpretation of the model is correct, if there's an extrinsic force imposing the orientation of the surface at every point, the model is not a serious contender to explain how an urchin is made. It's maybe less critical for other examples but leaves me unclear how seriously the possibilities illustrated by the model can be taken. That being said, the model certainly stands as a stimulant for further thinking along the same lines.

The above intuition about cell movement/rotation is correct as is now cooperated by Figure 2—figure supplement 3B. Also it is correct that our previous gastrulation Figure 7 used PCP to twist AB polarity and thereby enforce invagination from the outside. As our whole description of this process was too complicated we have changed to a simple one step scenario where all the gastrulation occur due to one particular set of initial conditions, and where invagination directly is caused by an initial twist of AB polarity of a subset of cells. We cannot claim that this a realistic scenario, but we propose it may be an indirect way of simulating the effects of bottle cells, and keep this example to illustrate the potential of our approach. Also we maintain the neurulation figure in supplement (Figure 1—figure supplement 11) although this still relies on the speculative assumption that changed PCP could direct movement of AB polarity. In any case the main role of PCP in our sea urchin gastrulation model is to direct the convergent extension after an initial deformation.

Similarly, in the case of fold formation on a sphere (Figure 5), which the authors suggest can occur if cell proliferation is on a faster time scale than relaxation, the physical origin of a slow relaxation time scale, and the mechanism of the observed instability, would merit discussion. Although assuming relaxational dynamics is standard and justified when the local minima of the energy are of interest, it might be questioned when turning to dynamical phenomena. Do folds develop through buckling, because the sphere grows (in number of cells) faster than it can expand in the surrounding medium? In many contexts, mechanical equilibration is much faster (seconds or minutes) than proliferation (hours). What would be the origin of a longer time scale for relaxation? Resistance from the surrounding medium might be one, but the authors present fast proliferation as an alternative mechanism to pressure from growth in a stiff gel.

**Author response image 2. respfig2:** At cell division, the daughter cell equilibrates by half a cell radius in one time unit, which is of order 1/1000 the generation time. Here, we show how a new cell (in blue) reaches equilibrium (in red). Cell division happens at time 0. At time 5, the next consecutive division happens in the system (not shown). The systems consists of 200 cells placed on a hollow sphere which is the initial condition in our organoid simulations (Figure 5). A new cell is introduced half a cell radius in a random direction away from the mother cell. The red line is the mean distance from the mother cell to all it’s neighbor cells at *t* = 1000. The generation time is intermediate (Figure 5A).

We appreciate the comment, which in part is associated to our poor description of the parameter *K* that should be divided by 10000 to become a growth rate. We have clarified this in the new Figure 5, and redefined the proliferation rate *K* as generation time *t_G_*. Buckling by growth in Figure 5 occurs when the doubling times of cells are between 140 and 4000 counted in our time units. From the potential in Figure 2 the typical displacement in one time unit is expected to be up to 0.5 cell radius (Figure 2, see also Author response image 2). Translated into a cell diameter of 10 micrometer and assuming that a cell generation of 1 day correspond to 1200 time units in the model, the cell movements correspond to up to 2 micrometer/minute.

A relatively fast local relaxation does not mean that the cells reach equilibrium at finite division rates, because the pressure from one division takes longer time to equilibrate on the longer distances that involves many cells. The entire system is driven out-of-equilibrium when the time between two consecutive divisions anywhere in the system becomes shorter than the relaxation time at the corresponding distances. Interestingly, by quantifying distribution of cell shapes it has been found that the growth of MDCK organoids is out-of-equilibrium and they implicated this to be a possible cause of organoid anisotropy (Cerruti et al., 2013).

I also have some issues of form with the manuscript (e.g. I do not find the connection that the authors draw between the model and evolution very substantive), but the above questions on the physical relevance of the model are the main issues that I would like to see clarified before the manuscript is given further consideration.

We agree, and have removed the evolution comment from the manuscript.

Reviewer #2:

1) This manuscript describes a mathematical model and associated simulations of large clusters of cells that can be assigned properties of apical basal polarity and planar cell polarity and may proliferate, to follow their 3D morphological evolution. The aim is to test the hypothesis that "stability, complexity, and diversity are emergent properties in populations of proliferating polarized cells." The authors conclude that "polarity enables complex shapes robust to noise but sensitive to changes in initial and boundary constraints." The work is of potential interest, but a number of substantial issues should be addressed. Most important, some essential definitions are not provided, the writing needs to be made more accessible to biologists who are not modelers, some key assumptions should be made explicit, and others.

We made numerous changes throughout the manuscript, as well as some additional supplementary figures that explain the way our potential acts (Figure 2—figure supplement 2 and Figure 2—figure supplement 3). See below for more details.

Essential revisions:2) Definitions needed: What precisely is meant by "stability?" Are the authors referring to feature constancy over time? Do they mean that models will converge to a specified set of features? What features? For example, they suggest that the trajectory in Figure 3 is stable, but noise seems to qualitatively change the shape (Figure 2—figure supplement 1), though it still produces a complex folded shape. It's not clear what, exactly, the authors are claiming.

Thank you for pointing this imprecision out. By stability we mean both constancy in time and ability to converge to similar sizes and relative position of lumens and folds in presence of noise. We realized that the reader may have been confused in comparing Figure 3D with Figure 2—figure supplement 1E–F. The simulations in Figure 2—figure supplement 1 and Figure 3 were started from different initial conditions and should not be compared for shape similarity.

The intention was to separately illustrate stability in time (in Figure 3) and stability to noise (in Figure 2—figure supplement 1E–F, where we compare end-results of the simulations run at high, Figure 2—figure supplement 1F, and low, Figure 2—figure supplement 1E, noise). The shapes emerging under high and low noise are not identical, however a high level of similarity – relative position and sizes of the majority of channels and lumens – is preserved. Ideally, this larger scale qualitative comparison, sketched in Figure 2—figure supplement 1E–F would benefit from a more rigorous quantification, by e.g. mapping changes in network of channels and lumens, but it is non-trivial and we judged it to be outside of the scope of this article. (To compare the relative impact of noise vs. initial conditions, Figure 2—figure supplement 1, we used distributions of pairwise distances between cells that were in the same position at start. This simple method however does not work for comparing macroscopical features).

We have rewritten the subsection”The final shapes are robust to noise but sensitive to initial and boundary conditions” describing stability results. We also modified Figure 2—figure supplement 1E–G to visually point to macroscopic similarities.

While addressing this point, we realized that the observed changes due to high noise come from noise acting only at early time, during “expansion” stage. The same level of noise applied after the system reached metastable state does not cause any major macroscopic changes, further supporting the link between polarity and morphological stability. We have added a sentence and Figure 2—figure supplement 1G to cover this point.

3) I struggled with the idea of "boundary conditions." I eventually concluded that, here, the term appears to mean restrictions on polarity orientations, and maybe other parameters. However, to a biologist, "boundary conditions" means something very different.

We have now introduced biological context for what we mean by boundary conditions and then defined the term “boundary condition” in subsection ”Folding by pressure or rapid proliferation result in different fold-morphologies”.

4) Difficulty with this passage was compounded by the challenging sentence "At the same time, in these symmetric cases the differences in initial conditions (no boundary conditions) are not sufficient to generate different structures." This implies that differences in initial conditions means no boundary conditions. I think the authors mean to say, "differences in initial conditions but without imposed boundary conditions."

Yes, this was the original meaning. We have adopted the suggested formulation.

5) Finally, in the Discussion section the authors say, "boundary conditions, e.g. fixed polarities for a fraction of the cells in some time." A precise definition should be provided when the idea of boundary conditions is first introduced.

We now define boundary conditions and we have also changed this sentence to: “We find that the exact morphological details are defined by initial conditions, e.g. initial positions and orientation of polarities, and boundary conditions, e.g. polarities restricted to certain direction for a fraction of the cells” in subsection ”Phenomenological description bridges cell polarity to macroscopic morphologies”.

6) "Minima" is defined in the Materials and methods section, but the authors also use "maxima." By analogy, one can guess what this means, but please define.

The “pronounced local maxima” was used to describe the folded structures. To simplify, we have chosen to reformulate this sentence to “As cells divide faster, our simulations predict a transition from a smooth spherical shell to an increasingly folded structure with multiple pronounced folds” in subsection ”Folding by pressure or rapid proliferation result in different fold-morphologies”.

7) Improve writing to make accessible to biologists: I'm a biologist with ample experience collaborating with modelers. Still, I found the manuscript difficult to follow and needed several readings to understand. If the goal is to make this work accessible to a broad audience (and it should be if published in eLife), the authors should carefully revise the text to replace jargon with precise simple language.

We have replaced a number of technical terms with a simpler expressions (e.g. replaced “metastable”, “topology”). Defined the terms “boundary conditions”, “stability” and tubes, and tried to simplify the language throughout.

8) Describe implications of model structure: Several features of the model are not as explicit as they should be. One is the conflation of PCP with convergent extension behavior. PCP defines a vectorial molecular asymmetry, and PCP can have a number of outcomes. One is CE, but others include the asymmetric elaboration of cellular structures, or the asymmetric induction of gene expression that results in different cell fates, for example. Here, the model structure equates PCP to forces that drive CE.

We agree. To clarify model implications, we have added a the following paragraph to subsection”Model implications”: “The present formulation of PCP effects has several implications. First, we restrict the effects of PCP to directed (anisotropic) cell–cell adhesion and do not consider its other possible roles, in e.g. asymmetric cell differentiation, thus primarily focusing on its role in CE. Second, in our current formulation AB polarity and PCP influence each others orientation on equal footing. However, as PCP is typically established in the apical plane, and is thus secondary to AB polarity, the current view is that AB polarity can influence PCP orientation and not the other way around.”

9) Furthermore, the descriptions of this implementation are misleading. In the Introduction, the authors, in describing the implementation of AB and PCP polarity, say "In other words, the adhesion strength between neighbor cells is modulated by the orientation of their polarities." Are the authors describing how AB and PCP (=CE) are implemented in the model, or how they function biologically?

To clarify, we reformulated the sentence to: “In other words, in our approach the adhesion strength between neighbor cells is modulated by the orientation of their polarities.”

10) There is no evidence that molecular PCP functions by regulation of adhesion (involvement of Celsr proteins notwithstanding), and differential adhesion is only one part of a complex and only partially understood set of mechanisms that drive CE.

We would like to thank the reviewer for the healthy dose of skepticism, as while initially this was also our view – originally in subsection”Cell–cell adhesion depends on the orientation of polarity” we stated “However, unlike with tight junctions, there is no biological confirmation for preferred directional adhesion with PCP” – this comment motivated us to do a careful literature survey.

We found that both proteins regulating adherens junctions, e.g. Smash (Beati et al., 2018) as well as adherence proteins, proteins forming adherens junctions, e.g. Bazooka, E-cadherins (Simões et al., 2010; Tamada et al., 2017; Levayer and Lecuit, 2013; Warrington et al., 2013; Aigouy and Le Bivic, 2016) are planar polarized. These, we believe, support our assumption of anisotropic, planar polarized adhesion. In case, if the arguments below are notwithstanding, we would very much appreciate reviewer's opinion on where they fail.

We have now modified the paragraph in subsection”Cell–cell adhesion depends on the orientation of polarity”, below Equation 8, which now reads “Cells adhere to each other by membrane proteins assembled in adherens junctions just below the apical surface. Both proteins regulating adherens junctions, e.g. Smash (Beati et al., 2018) as well as adherence proteins forming adherens junctions can be planar polarized, e.g. Bazooka, E-cadherins (Simões et al., 2010; Tamada et al., 2017; Levayer and Lecuit, 2013; Warrington et al., 2013; Aigouy and Le Bivic, 2016). These indirectly support our assumption of anisotropic, planar polarized adhesion.“

We do agree that likely there are other mechanisms contributing to convergent extension, and have added this to the DDiscussion section which now reads “The presented formulations of model captures only some of the known events contributing to CE and does not include PCP-driven cell-shape changes mediated by e.g. apical constriction (Nishimura, Honda, and Takeichi, 2012) or contribution of external forces (Lye and Sanson, 2011).”

11) Another implicit part of the model as I understand it, and one that is probably very important to the outcomes, is the notion that altering PCP vectors can influence the AB vectors of cells. I know of no biological precedent to justify this behavior. This input assumption is inconsistent with my understanding of the biological reality. AB polarity is an initial condition that must be met for the subsequent establishment of PCP, which is then by definition orthogonal to the AB axis. There are examples in which PCP determines edges of groups of cells that then increase tension, thereby inducing folds, but this is a different mechanism from the one implied here.

We again appreciate the input, and have shown that constraining PCP to the apical plane (more details on this in our reply below) does not affect CE and formation of tubes neither when we start form spherical lumen (Figure 6—figure supplement 1) or simulate sea urchin gastrulation.

12) I think it would be interesting for the authors to explore how their model would operate if AB polarities orient PCP, but PCP cannot directly influence AB. I suspect that the examples of tube formation would fail. At a minimum, they should discuss the need for this assumption explicitly, noting that it is not currently supported by biological data (or citing the data if there is some that I'm not aware of).

Thank you for stressing this point. Indeed, in our formulation (described by Equation 7) PCP and AB polarity can influence each on equal footing, and we used this symmetry to propose an alternative hypothesis for invagination during sea urchin gastrulation (orientation of PCP set at the boundaries drive the orientation of AB polarity).

While we did explore the consequences of unidirectional, AB polarity to PCP, effects, we also found data supporting that PCP may be primary to, and direct AB polarity.

Consequences of constraining PCP: We have explored the consequences of constraining the hierarchy or polarities in two ways: First, in case of sea urchin gastrulation, without modifying our model we impose boundary conditions directly onto AB vectors (Figure 7; the entire section on sea urchin gastrulation was updated to reflect this change). Biologically, this is an implicit way to capture the effect of the bottle cells, where the change in cell shapes effectively reorients AB vectors in neighboring cells (see Figure 2—figure supplement 2). Second, we have directly tested how disabling PCP pull on AB polarity (setting λ_2_ to 0 when updating Equation 10) affects tube formation. We observe no qualitative difference compared to our original results in neither tube formation (Figure 6—figure supplement 1), nor sea urchin gastrulation (not included), thus suggesting that we could relax our assumption symmetry between PCP and AB influence.

Data supporting that PCP may influence AB polarity: The idea that AB polarity and PCP may act on equal footing is appealing both for its symmetry and simplicity, and we were excited to find a number of recent data supporting that PCP is not necessarily secondary to AB polarity and may regulate AB polarity and influence its orientation.

i) Cells can acquire PCP without AB polarity present. The two examples are the radial intercalation of the multiciliated cell progenitors in *Xenopus* epithelia (Ossipova et al., 2015) and transition from the solid cord to a tube in intestine morphogenesis of Zebrafish (Baer, Chanut-Delalande and Affolter, 2009; Zorn and Wells, 2009). In both of these examples, cells acquire AB polarity after convergent extension / intercalation events driven by PCP.

ii) Proteins required for AB polarity can become planar-polarized. These include: Baz/Par-3 protein (Warrington, Strutt and Strutt, 2013; Aigouy and Le Bivic, 2016) which in *Drosophila* development has a role in both AB polarity and PCP (Beati et al., 2018); PCP regulates Lgl, protein central for establishing AB polarity (Choi and Sokol, 2009; Dollar et al., 2005) and Lgl as well as other AB polarity proteins are planar polarized in *Drosophila* epidermis (Kaplan and Tolwinski, 2010).

iii) The changes in cell-shapes during invagination (e.g. sliding of adherens junctions and formation of bottle cells) are regulated by PCP. These changes in cell-shapes effectively reorient AB polarity (see Figure 2—figure supplement 2). Apical constriction leading to formation of bottle cells is under PCP control during neural tube closure (Ossipova et al., 2014; Nishimura, Honda and Takeichi, 2012; Kinoshita et al., 2008); gastrulation in *C. elegans* (Lee et al., 2006), sea urchin (Croce et al., 2006) and *Xenopus* (Choi and Sokol, 2009). Also, the sliding of adherens junctions leads to shorter AB axis and is mediated by a Baz (key protein in PCP and AB polarity) during *Drosophila* gastrulation (Wang et al., 2012). Thus, the data shows that PCP can drive changes of the cell-shape (and consequently orientation of AB polarity) and while does not directly explain how the orientations of PCP and AB polarity are coordinated, it supports our assumption that components of PCP pathway can influence the orientation of AB polarity.

These points are now included in the subsection ”Model implications”.

13) Some unlikely assumptions: While it may be interesting that complex forms such as the sea urchin-like gastrulation and the folded neural tube CAN be produced within the confines of the model, the conditions needed stretch the bounds of plausibility. The model simulating sea urchin gastrulation requires a precise choreography of PCP parameters that could happen but seem unlikely. Even more difficult to believe is the initial conditions invoked to simulate neural plate folding. Specifically, invoking "two rows where the PCP points parallel and antiparallel to AB polarity" invokes conditions for which I know of no precedent at all.

We should have been more explicit on how our assumptions for both cases relate to the current understanding of these processes. In case of sea urchin gastrulation, the process is believed to consist of two stages, primary invagination driven by swelling of the inner layer of extracellular matrix beneath invaginating cells (Lane et al., 1993) and formation of a ring of bottle cells due to apical constriction (Kimberly and Hardin, 1998) and secondary invagination were tube elongates due to CE and pull by mesenchymal cell (Lyons, Kaltenbach, and McClay, 2012). This was the motivation for two discrete stages in our original model formulation.

We have now converged to a simpler, less choreographed version, where all conditions can be set prior to invagination and PCP remains in apical plane. First, we set boundary condition such that planar cell polarities of the invaginating cells are oriented on a spiral, and are all in the apical plane. This constraint allows for convergent extension and is present at all times. While not documented, it is consistent with observed effects of WNT orienting PCP within apical plane (Humphries and Mlodzik, 2017). Second, instead of driving invagination by PCP orientation, we simulate the effect of bottle cells (see Figure 2—figure supplement 2) by reorienting AB polarities. This external force on AB polarity can be thought of as a combined effect of bottle cells and bending by swelling of neighboring extracellular matrix. Motivated by the observations in other species where the changes of cell shapes are sufficient for invagination (does not require swelling of neighboring extracellular matrix), we have tested the case where the AB polarities of bottle cell neighbors prefer to be tilted towards each other (as in Figure 2—figure supplement 2). This completely cell-intrinsic scenario produces similar invagination outcome.

These changes are now described in a new Figure 7 and updated subsection ”Phenomenological description bridges cell polarity to macroscopic morphologies”.

The most provoking and least likely assumptions in our initial formulation was the hypothesis that boundary conditions may set PCP to point out of the apical plane. We formulated it as direct effect of instructive cues by organizing morphogens (e.g. WNT), however this effect may be indirect and act through e.g. reported cell-shape changes where PCP drives apical constriction (see our reply to point 12), which tilts AB axis in neighboring cells, thus tilting PCP orientation.

In case of neurulation, the idea was to simulate the neural plate (cells in the middle, between the two rows with constrained PCP) surrounded by the epidermis (the rest of cells). The two rows of cells with PCP pointing out of epithelial plane would then correspond to the cells at the dorsolateral hinge points next to the neural plate (epidermis boundaries). In chick spinal neural tube can close with only these two hinge points (Nikolopoulou et al., 2017). The bending is driven by apical constriction and PCP is essential for bending, CE and closure (Nikolopoulou et al., 2017). We used very simple brute-force constrain on PCP (pointing out of the epithelial plane) at the boundaries, meant to represent hinge points, to show that this could drive the entire process. Here again, this may be an indirect effect of PCP → Apical constriction → tilt of AB polarity → PCP points out of the original plane of epithelium. Including an intermediate step linking PCP with apical constriction (by e.g. introducing potential favoring tilted AB polarities) and convergent extension is rather simple and would likely result in a more robust model and require less fine-tuning of boundary conditions but is beyond the scope of the current manuscript.

We have updated the Discussion section to include our reflections on PCP pointing out of plane.

14) Additional discussion of testable predictions needed: Re-emergence of folds in pancreatic organoids, pressure causes fewer and much deeper folds that form early during growth. This seems to be a qualitative measure. How would biological observations be assigned to one or the other category?

Thank you! We realized that we did not stress the quantitative aspect well enough and have edited the Discussion section.

In addition to the qualitative description, our results also suggest that the two scenarios can be distinguished quantitatively by the differences in depth and length of the folds (new Figure 5—figure supplement 1). We now refer to this figure also in the Discussion section.

15) Isn't the key difference that the pressure hypothesis predicts that the number of minima maxes out, whereas the proliferation model predicts that the number keeps growing with increase in size (Figure 5)?

Yes, it is indeed a key difference and is now clarified both in the Results section and in the Discussion section.

16) Re tubes formed by nonproliferating cells, "in PCP mutants with shorter and wider tubes one would expect higher fraction of "AB adhesion" relative to "PCP adhesion". However, as noted above, PCP is not directly related to any known adhesion events.

We have now elaborated on this prediction in the Discussion section. Please also see our reply to point 9 above. We also realized that our formulation “PCP adhesion “was probably confusing. We did not mean to claim this has to be a direct effect of PCP. Because we represent cells as point particles, the PCP in our model is a phenomenological description and is meant to represent the combined effect of anisotropic, both extra- and intracellular, factors (e.g. anisotropic apical constriction, polarized junctional actomyosin) that would direct cell movements and favor its position among other cells depending on orientation of these factors.

17) Re the conditions differentiating between tubes forming perpendicular (e.g. sea urchin gastrulation) or parallel (as in Drosophila gastrulation or neurulation) to the plane of epithelium, the measurements could in principle be made, but as noted above, the conditions seem to me to be highly implausible, and ask us to ignore the likely contribution of other signals.

We have now added an extensive discussion addressing both the implausibility of PCP pointing out of plane of epithelium as well as the contribution of other factors (Discussion section).

Reviewer #3:

1) This is a creative and innovative article on development. The authors introduce a simple theoretical model based on the idea that apical-basal (AB) polarity and planar cell polarity (PCP) are (the) main driving forces relevant for the formation of complex tissue morphologies. The model idealises cells as point particles with two independent axes (polarities), an AB and a PCP axis. Pairwise interactions between cells contain an attractive and a repulsive component where the magnitude of the attractive component depends on three additive factors: alignment of the AB axis, alignment between AB and PCP axis, and alignment of PCP axis. Noise comes into the model at various levels: Langevin noise in the equations of motion, noise at the level of choosing the axes orientation at cell proliferation, and noise at the level of initial conditions. In summary, this seems to be a computationally efficient model to explore the question: How can cell-cell adhesion that depends on the relative orientation of polarisation axes alone drive different morphologies during development?Indeed, the authors simulations show that their model is able to reproduce a broad range of morphologies depending on several factors: choice of initial conditions, relative tendencies of the various axis to orient with respect to each other, and cell proliferation rate. A clear strength of the computational model (I would not call it a theory) is that it provides a mechanistic link between morphology and cell-cell interaction as well as initial conditions. Thus, I find it a clear conceptual advance in the field which puts a fresh perspective on the role of generic interactions for the emergence of different tissue morphologies.

We agree “theory” may overstate the nature of our work. The main motivation for calling it theory was to highlight that our approach is intentionally phenomenological and can work with few parameters. Computational model is a broad term and often associated with dozens of parameters. We have compromised the two by replacing “Theory” in the title with “Theoretical tool”.

2) The authors also list a few testable predictions. Here, I am not so convinced yet for very basic reasons: Testable predictions require either a clear link between a molecular and an emergent process, or a set of mutually exclusive alternatives that can be tested.

We agree that by its nature our phenomenological description is outside the realm of mechanistic predictions targeted at specific proteins. While the alternatives we consider are not mutually exclusive, one can in principle test how much each contributes to the final result e.g. varying the stiffness of the gel and generation time. We now added a more detailed discussion on this in the Discussion section.

The scenarios explored for pancreatic organoid folding are not limited to only two possibilities mentioned in the paper. (Other possibilities are the instabilities associated to secreted morphogens and the bias of cell differentiating into slow and fast growing cells depending on their position in the fold.) We see our model and predictions as a starting point and motivation for quantitatively testing organoid morphologies as our results suggest that different scenarios may result in quantitatively different shapes. The model will be rather easy to modify to either consider other alternatives or explore the combined effect of several, but we lack constraints by experimental data for this to be meaningful.

3) Concerning the latter case (in the section of organoids) the authors analyse two possible scenarios (cell proliferation and a specific way to implement external forces). These are not two mutually exclusive alternatives but only two out of many possible conceivable scenarios. Concerning the former case, in the example on sea urchin gastrulation the authors seems to a large degree put in what they would like to get out.

In regards to gastrulation, we have simplified the modeling of gastrulation greatly in the new version, reducing it to one step: one initial condition of twisted AB polarities and one boundary condition in form of an external constraint/forcing on PCP. This emphasizes the simple interplay of initial invagination and convergent extension, and makes the connection between input and output even more transparent. We very much agree that a more realistic approach should include forces that favour invagination from cell shapes (e.g. bottle cells) and various checkpoints and differentiation processes along the developmental pathway.

4) Finally, and this is more on the side of a personal opinion, I am not convinced that morphologies that strongly depend on initial conditions are a robust way for cellular development. For instance, how can an organism make sure that all these initial conditions are properly chosen?

Agree, but in Figure 2 we did not have any external constraints (boundary conditions). Overall, we try to convene that both initial conditions and external constraints are important. Initial conditions can be overruled by boundary conditions, i.e. persistent constraints on cell polarities by e.g. neighboring tissues through intra- and extracellular feedbacks. The robustness of the developmental process most likely stems from these constraints. All the finally obtained morphologies are subsequently robust to noise, growth and disruptions (see also the new Figure 3—figure supplement 2).

5) In summary, I found the paper a well-written, creative, and thought-provoking contribution to the field of development. It puts the emphasis on local and rather generic interactions between cells' polarities. If the authors can convincingly reply to my above queries, I would be inclined to recommend it for publication.

We are very happy for this overall positive view on our work, and hope that our response and the resulting changes, including in particular our simpler gastrulation treatment and more modest claims, sufficiently improved the manuscript.